# SELF-SUPERVISED DYNAMICAL SYSTEM REPRESENTATIONS FOR PHYSIOLOGICAL TIME-SERIES

## ABSTRACT

The effectiveness of self-supervised learning (SSL) for physiological time series depends on the ability of a pretraining objective to preserve information about the underlying physiological state while filtering out unrelated noise. However, existing strategies are limited due to reliance on heuristic principles or poorly constrained generative tasks. To address this limitation, we propose a pretraining framework that exploits the information structure of a dynamical systems generative model across multiple time-series. This framework reveals our key insight that class identity can be efficiently captured by extracting information about the generative variables related to the system parameters shared across similar time series samples, while noise unique to individual samples should be discarded. Building on this insight, we propose PULSE, a cross-reconstruction-based pretraining objective for physiological time series datasets that explicitly extracts system information while discarding non-transferrable sample-specific ones. We establish theory that provides sufficient conditions for the system information to be recovered, and empirically validate it using a synthetic dynamical systems experiment. Furthermore, we apply our method to diverse real-world datasets, demonstrating that PULSE learns representations that can broadly distinguish semantic classes, increase label efficiency, and improve transfer learning.

## 1 INTRODUCTION

Self-supervised learning (SSL) is a powerful framework for learning general-purpose representations from unlabeled datasets. These representations may be especially powerful for physiological time-series, making it possible to track physiological states, detect diseases, and improve our understanding of biology (Perochon et al.; Chen et al., 2023b; Li et al., 2025a). The success of an SSL approach here is determined by how well its pretraining objective guides an encoder to preserve information relevant to the identity of the underlying physiological process while filtering out irrelevant noise (Tian et al., 2020).

Although many time-series SSL methods have been proposed, their strategies often rely on heuristic assumptions about which information the model should extract, which may not align well with the characteristics of physiological signals. For instance, time-series contrastive learning (CL) aims to learn representations that encode information shared between two views of the data while remaining invariant to unshared information. However, positive pairs formed through augmentations (e.g. jittering, scaling) can incorrectly relate samples with different clinical diagnoses, and sampling heuristics (e.g. temporal-proximity) may fail to capture nonstationarity, where physiological states may shift abruptly over time. Masked autoencoding (MAE) takes a different approach, using a reconstruction task to recover information shared between masked and unmasked regions while ignoring unshared features (Kong and Zhang, 2023). Yet, similar to CL, masking strategies (e.g., random, patch, channel, frequency) are heuristically designed without regard to the data-generating process, which can inadvertently treat important temporal relationships as noise, thereby misrepresenting the underlying process. Heuristically designed modeling choices often do not explicitly state the assumptions about the data, identify which sources of information are captured, or explain how the method targets those sources, and are primarily selected to empirically improve downstream performance. Consequently, SSL methods that rely on heuristics may fail to preserve information essential for distinguishing physiological states which limits their effectiveness when applied to real-world clinical tasks (Xiao et al., 2020).

Figure 1: Intuition behind PULSE. A dynamical systems model of a physiological time-series dataset allows us to distinguish between information that is shared between similar time series and information that is sample-specific and not transferrable. PULSE leverages this distinction to learn representations that preserve shared system information while discarding sample-specific ones.

Sequential variational autoencoders (SVAEs) offer an alternative to heuristic SSL by defining an autoencoding task based on the characteristics of physiological time-series. Specifically, SVAEs explicitly define the type of information to be extracted via a dynamical systems generative model and use structured decoders to guide learned representations toward components of this process. By defining a generative model, assumptions about the data are made clear which allows SVAEs to prioritize the recovery of specific sources of information about time-series dynamics that heuristic methods may ignore or distort. This makes them broadly effective for physiological time-series, where information about the dynamics drive the signal's evolution and directly reflect the underlying state (Nayak et al., 2018; Sherman, 2011; Marmarelis, 2004; McKenna et al., 1994; Brooks et al., 2021).

However, SVAEs have two critical limitations that prevent them from consistently outperforming heuristic SSL methods. First, the autoencoding task does not separate signal from noise, as both are reconstructed jointly. For physiological time series, this distinction is critical because not all observed information reflects the underlying state. For example, the initial time or amplitude of a recording is much less informative of a signal's identity than its temporal behavior, and transient fluctuations often occur naturally without indicating changes in the physiological process. Second, because samples are reconstructed independently, there is no mechanism to learn information shared across different samples. Since physiological time series often exhibit repeated patterns, capturing this shared information is important because it can relate different samples with similar behaviors.

In this work, we address the limitations of heuristic SSL and dynamical systems models for physiological time-series by introducing **PULSE** (**P**hysiological self-s**U**pervised **L**earning using **S**ystem **E**ncoders). This is accomplished by introducing a novel pretraining objective that selectively retains information relevant to the underlying process based on structure available from a dynamical systems generative model. In our framework, we conceptualize a time series dataset as the result of a hierarchical graphical model where related samples are generated from the same underlying dynamical system. Under this model, we obtain our key insight as illustrated in Figure 1: *system information* from the generative parameters should be preserved since it relates samples produced by similar processes, whereas *sample-specific information* about factors that are unique to each sample, such as initial conditions and process noise, is non-transferrable and should be discarded. To operationalize this idea, we propose a practical cross-reconstruction task that encourages the learned system representation to be invariant to sample-specific factors by reconstructing multiple similar time series across randomly sampled initial conditions. We then present theory on sufficient conditions under which system information is recovered and empirically validate this theory through synthetic experiments. Finally, in several real-world datasets and tasks, we demonstrate that PULSE achieves consistent performance improvements over recent CL and MAE methods. Our contributions are summarized as follows:

1. This is the first work to develop an SSL pretraining approach that uses the information structure in dynamical systems to explicitly separate relevant and irrelevant information. We formalize this distinction through a hierarchical generative model over a time-series dataset and propose a practical pretraining strategy that selectively extracts the desired information.
2. We explore theory for PULSE that could provide conditions for when system information is recovered and empirically validate these ideas with a synthetic dynamical systems experiment.
3. In many real-world physiological applications, PULSE achieves SOTA performance, outperforming a representative set of baselines in linear evaluation, data efficiency, and transferability.

## 2 BACKGROUND AND RELATED WORK

**Self-Supervised Learning for Time Series.** CL and MAE are the most studied SSL paradigms for time-series due to their success in computer vision (Gui et al., 2024; Li et al., 2024; 2025b). In CL, positive pair design is crucial and determines what information the representation is invariant to (Tian et al., 2020). For time series, positive pairs created via augmentations like scaling, jittering, or adding noise as in SimCLR (Chen et al., 2020) often yield inconsistent performance across datasets (Liu et al., 2024). Positive pairs can also be constructed via sampling methods such as selecting time neighbors in TNC (Tonekaboni et al., 2021), creating two crops of a single segment in TS2vec (Yue et al., 2022), or through a learned reconstruction measure as in REBAR (Xu et al., 2023). In MAE, the masking strategy plays a central role in determining what shared information the model must recover to solve the masked reconstruction task (Kong and Zhang, 2023). TimeMAE (Cheng et al., 2023) introduces block masking to capture shared information across sub-series. PatchTST (Nie et al., 2022) applies a similar idea termed patch masking, and further shows that assuming channel independence can improve forecasting performance. Wang et al. (2024) explores various time and channel masking strategies, but finds their performance to be highly dataset-dependent. CiTrus (Geenjaar and Lu, 2025) proposes a MAE task that predicts either spectrogram or time-series, but shows that neither approach is consistently effective across a range of biosignals. Sensitivity to dataset characteristics limits these approaches when the dataset properties are not well understood a priori. Instead, PULSE is based on a dynamical systems model across multiple time series and demonstrates consistent performance in many domains.

**Dynamical Systems Models of Physiological Time Series.** Many physiological time-series arise from underlying physical and biochemical processes that can be well described by dynamical systems models. Thus, dynamical systems provides a shared modeling framework useful for describing a broad range of physiological processes. Since a physiological system evolves independently of how it is measured, the observations $y_t = g_y(x_t)$ can be viewed as the output of an observation function applied to a latent physiological state $x_t \in \mathbb{R}^n$, which itself evolves over time according to the dynamics $x_{t+1} = g_x(x_t, \Theta)$, where $g_x$ is parameterized by system variables $\Theta$. This defines a generative process known as the *state-space model* (SSM). In this framework, a time series is generated by specifying an initial condition $x_{t_0}$, evolving it forward in time according to $g_x$, and producing measurements according to $g_y$. Although dynamical systems models have been successfully used to study a wide range of physiological processes, including cardiac dynamics (Bianco et al., 2024), brain activity (Chen et al., 2024; Mudrik et al., 2024), and motor control (Shenoy et al., 2011), they have not yet been used to guide which information should be preserved or discarded in a SSL objective. In this work, we demonstrate that exploiting dynamical system structure in time-series SSL pretraining can lead to consistent improvements in representation learning for physiological signals.

**Sequential Variational Autoencoders.** SVAEs can be viewed as a time-series SSL strategy that leverages the state-space formulation to learn representations corresponding to components of a dynamical system models. Training is performed through an autoencoding task by maximizing the ELBO, $\mathbb{E}_{q(z|y)}[\log p(y|z)] - \beta \mathrm{KL}(q(z|y)\|p(z))$, where $z$ are latent variables, $q(z|y)$ is the approximate posterior, $p(y|z)$ is the likelihood, $p(z)$ is the latent prior, and $\beta$ is the regularization strength for the KL term. The encoder $q(z|y)$ and decoder $p(y|z)$ are often parameterized by neural networks, and are structured to reflect specific generative assumptions. For instance, LFADS (Sussillo et al., 2016) encodes each time series into an initial condition $x_{t_0}$ that evolves under a shared dynamics function and uses a GRU decoder to reconstruct the observed data. DSVAE (Li and Mandt, 2018) assumes that the dynamics is data-dependent and can be factorized into static and dynamic generative factors. Unlike CL and MAE, which aims to selectively recover useful sources of information during pretraining, SVAE methods share the common limitation that the $\mathbb{E}_{q(z|y)}[\log p(y|z)]$ term does not distinguish between meaningful signal from noise and instead encourages explaining all observed variability in the data. Consequently, the learned representations can be overly sensitive to irrelevant information, reducing their transferability to downstream tasks.

## 3 PULSE APPROACH

In this section, we present PULSE. First, we develop a dynamical systems generative model for multiple time-series samples to identify which information is transferrable and which is not. Then, we introduce a practical pretraining strategy to extract the transferrable information. Finally, we provide theoretical guarantees for conditions under which the transferrable information is recovered.

### 3.1 Revealing Shared Information Between Multiple Time-Series

To determine which information should be retained or discarded, we first need to characterize the information present in a physiological time-series dataset. We do this by introducing notation to describe the data and constructing a graphical model of its generative process, which makes explicit the sources of information available in the observations.

**Notation.** A physiological time-series dataset often consists of long continuous time series recordings that are segmented into shorter, fixed-length windows. Let $\mathbf{D} \in \mathbb{R}^{R \times T \times M}$ denote a dataset of $R$ recordings, each consisting of $T$ time steps and $M$ measurement channels. We use $\mathbf{D}_{r,\tau:h}$ to denote a subsequence from the $r$th recording, starting at time index $\tau$ and ending at index $h$. We can then construct a dataset of $N$ time-series windows, $\mathbf{Y} \in \mathbb{R}^{N \times W \times M}$, where each window is defined as $\mathbf{Y}_i = \mathbf{D}_{r_i, \tau_i : \tau_i + W}$ for a window size $W$, with $r_i$ and $\tau_i$ denoting the recording and start index of the $i$th sample, respectively. A specific time-slice is denoted as $\mathbf{Y}_{i,t_1}$, where $t_1 \in \{1, \dots, W\}$, and consecutive time-slices are defined as $t_{k+1} = t_k + 1$, where $k$ counts the steps forward from $t_1$.

**Time-Series Dataset Generative Model.** As shown in Figure 2, we model $\mathbf{Y}$ with a dynamical systems generative model, where each $\mathbf{Y}_i$ is produced by a latent system with system parameters $\mathbf{\Theta}_i$, and an initial condition $\mathbf{X}_{i,t_0}$. While each sample could in principle be generated by a unique system, physiological activity is often stereotyped, with many underlying processes exhibiting consistent, repeatable patterns over time. For example, a walk cycle captured by an accelerometer displays repeated phases such as heel strike, mid-stance, and toe-off, while an ECG signal shows recurring PQRST complexes during normal sinus rhythm. As a result, different samples may share the same underlying system, and the number of unique $\mathbf{\Theta}_i$ is generally smaller than $N$. To capture this, we define $\mathcal{I}_s$ as the set of indices for samples generated by system $s \in \{1, \dots, S\}$, and $\mathbf{\Theta}^{(s)}$ as the system parameters shared across all samples in $\mathcal{I}_s$. In other words, $\mathbf{\Theta}_i = \mathbf{\Theta}^{(s)}$ for all $i \in \mathcal{I}_s$. The joint distribution for this generative model is given by,

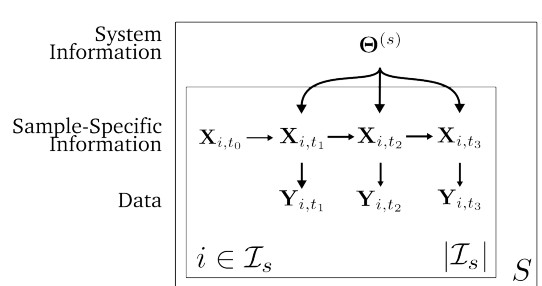

Figure 2: Our graphical model of multiple time-series windows, based on dynamical systems, distinguishes transferable system information shared across similar time-series from non-transferable information unique to each sample such as initial conditions and process noise.

$$p(\mathbf{Y}, \mathbf{X}, \mathbf{\Theta}) = \prod_{s=1}^{S} \prod_{i \in \mathcal{I}_s} p(\mathbf{X}_{i,t_0}, \mathbf{\Theta}^{(s)}) \left[ \prod_{k=1}^{W} p(\mathbf{Y}_{i,t_k} | \mathbf{X}_{i,t_k}) \right] \left[ \prod_{k=2}^{W} p(\mathbf{X}_{i,t_k} | \mathbf{X}_{i,t_{k-1}}, \mathbf{\Theta}^{(s)}) \right], \quad (1)$$

where $\mathbf{X}_{i,t_0}$ and $\mathbf{\Theta}^{(s)}$ are independent. Here, the observation density $p(\mathbf{Y}_{i,t_k} | \mathbf{X}_{i,t_k})$ is defined by the SSM measurement equation $\mathbf{Y}_{i,t_k} = g_y(\mathbf{X}_{i,t_k}) + \epsilon_{i,t_k}$ and the transition density $p(\mathbf{X}_{i,t_k} | \mathbf{X}_{i,t_{k-1}}, \mathbf{\Theta}^{(s)})$ is defined by the SSM transition equation $\mathbf{X}_{i,t_k} = g_x(\mathbf{X}_{i,t_{k-1}}, \mathbf{\Theta}^{(s)}) + \nu_{i,t_k}$, where $\epsilon_{i,t_k}$ and $\nu_{i,t_k}$ are noise terms.

The factorization in Eq. 1 reveals a hierarchy of information in $\mathbf{Y}$ that is also illustrated in Figure 2. One source of information comes from the system variables $\mathbf{\Theta}^{(s)}$, which govern the evolution of the time series and are shared across all $\mathbf{Y}_i$ generated by the same system. Since $\mathbf{\Theta}^{(s)}$ is shared across all samples in $\mathcal{I}_s$, it provides information that is transferable across different samples. This observation leads to the following notion of similarity between independent time-series samples.

**Definition 3.1** (Similar Time-Series). Two time-series $\mathbf{Y}_i$ and $\mathbf{Y}_j$ are similar if they are generated by the same system parameters $\mathbf{\Theta}^{(s)}$, such that both indices satisfy $i, j \in \mathcal{I}_s$.

In other words, two time series are considered similar when they share the same underlying dynamics information, while any differences arise solely from their sample-specific components. Learning a representation that captures this system information produces a space in which samples with similar dynamics are naturally grouped together.

Figure 2 also illustrates sample-specific sources of information that is unique to each $\mathbf{Y}_i$, such as the initial value $\mathbf{X}_{i,t_0}$, as well as observation and dynamics noise $\epsilon$ and $\nu$. In our setting, a

representation should be invariant to this information, as it is not shared across $\mathbf{Y}_i$ and cannot be transferred between different samples. This insight is especially relevant for physiological time series, where a signal's identity is determined by the underlying system dynamics rather than by the exact starting value, sensor noise, or transient fluctuations.

Therefore, *by extracting system information and discarding sample-specific information, we obtain a representation that can relate physiological time series based on their temporal characteristics while ignoring irrelevant factors.*

### 3.2 PULSE PRETRAINING FOR RECOVERING SYSTEM INFORMATION

Our goal is to design a practical pretraining strategy that encourages an encoder to recover system information while ignoring sample-specific factors. A promising strategy is a dynamical systems *cross-reconstruction* task, where given two samples from the same system (i.e., $\mathbf{Y}_i$ and $\mathbf{Y}_j$ with $i, j \in \mathcal{I}_s$), the system information inferred from $\mathbf{Y}_i$ is used to reconstruct an independently realized sample $\mathbf{Y}_j$. By requiring the system representation from $\mathbf{Y}_i$ to reconstruct multiple random samples of $\mathbf{Y}_j$, an encoder is encouraged to keep only the shared information between these samples. According to Eq. 1, the only shared variables are $\mathbf{\Theta}^{(s)}$, and encoding irrelevant factors may lead to poor reconstruction.

**Cross-Reconstruction with Similar Pairs.** To formalize this approach, we define an inference step and generative process given sample pairs $(\mathbf{Y}_i, \mathbf{Y}_j)$ produced by $\mathbf{\Theta}^{(s)}$ as input. For inference, we introduce two encoders that separate transferable from non-transferable dynamical systems components: a system encoder $f_{\text{sys}}$ to estimate shared system information and an initial condition encoder $f_{\text{init}}$ to estimate the sample-specific initial condition. As shown in Figure 3, $f_{\text{sys}}$ uses the dilated convolution (Yue et al., 2022) to extract information across the entire window according to $\mathbf{\Theta}_i = f_{\text{sys}}(\mathbf{Y}_i)$. In contrast, $f_{\text{init}}$ is implemented as a 2-layer CNN whose receptive field is centered around a specified time $t_0$, producing $\mathbf{X}_{j,t_0} = [f_{\text{init}}(\mathbf{Y}_j)]_{t_0}$ where $t_0 = 1$ selects the initial condition needed to reconstruct a sample from the first time step until the end of the window. For generation, we use an SSM decoder and define the cross-reconstruction objective,

$$\mathcal{L}_{\text{Cross}}(\mathbf{Y}_i, \mathbf{Y}_j) = \mathbb{E}_{(\mathbf{Y}_i, \mathbf{Y}_j) \sim \mathcal{P}} \left[ \sum_{k=1}^{W} \|\mathbf{Y}_{j,t_k} - g_y(g_x(\mathbf{X}_{j,t_{k-1}}, \mathbf{\Theta}_{i,t_k}))\|^2 \right], \quad (2)$$

where $\mathcal{P}$ is a distribution over sample pairs $i, j \in \mathcal{I}_s$, and $g_x$ and $g_y$ are parameterized by a GRU and linear projection layer respectively. Here, we implement $\mathbf{\Theta}_i$ as the GRU input, since its hidden state evolves according to dynamics defined by input-dependent gating. Additionally, following prior dynamical systems methods (Li and Mandt, 2018), we further factorize $\mathbf{\Theta}_i$ into separate time-invariant and time-varying components to represent nonstationary behaviors, a key feature of physiological time-series. To do this, we decompose $\mathbf{\Theta}_i$ into a time-invariant component $\boldsymbol{\theta}_i$, obtained via max pooling over time, and a time-varying component $\tilde{\boldsymbol{\theta}}_{i,t_k}$, obtained via a two-layer CNN over the latent dimension, and then concatenate the result at each time step to form $\mathbf{\Theta}_{i,t_k} = [\boldsymbol{\theta}_i, \tilde{\boldsymbol{\theta}}_{i,t_k}]$. Importantly, as shown in equation 2, $\mathbf{\Theta}_{i,t_k}$ does not include parameters for $g_y$ and only includes parameters for $g_x$. This design choice is motivated by the SSM formulation, where the parameters of the observation function are not considered part of the underlying dynamics, reflecting the idea that how a process is measured is separate from the dynamics of the process itself. Therefore, the resulting embedding for $\mathbf{\Theta}_{i,t_k}$ includes only the pooled output of the dilated convolution.

To minimize Eq. 2, $f_{\text{sys}}$ must extract shared information from $\mathbf{Y}_i$ that can explain the evolution of $\mathbf{Y}_j$. This means encoding only the underlying system variables $\mathbf{\Theta}^{(s)}$, since these are the factors that remain invariant across samples from the same system. Furthermore, because $\mathcal{P}$ involves optimizing over randomly chosen pairs, $f_{\text{sys}}$ cannot rely on sample-specific information from $\mathbf{Y}_i$, since this information will not be present in $\mathbf{Y}_j$ and may increase $\mathcal{L}_{\text{Cross}}$ if present. Thus, the $\mathcal{L}_{\text{Cross}}$ loss encourages the encoder to discard non-shared factors and focus only on shared system information.

**Cross-Reconstruction with PULSE Pseudo-Pairs.** Unfortunately, a problem in Eq. 2 is that, in an unlabeled dataset, we do not have access to $\mathcal{I}_s$ and therefore cannot sample from $\mathcal{P}$. Instead, we propose a sampling strategy that constructs semantically similar time series pseudo-pairs $(\mathbf{Y}_i, \widetilde{\mathbf{Y}}_i)$ from a single sample. Since $\mathbf{\Theta}^{(s)}$ are independent of the initial condition in Eq. 1 and should define dynamics that can generate time series from any starting state, representations that capture system information should generalize across multiple initial conditions.

However, initial conditions cannot be chosen arbitrarily, as data may not be available to supervise reconstruction from arbitrary starting points. To address this, we select an initial condition at a random time step $t_0 \sim$ Uniform$(1, W/2)$ within each window $\mathbf{Y}_i$, and task the model with reconstructing the segment from $t_0$ to the end. Note we restrict $t_0$ to the first half of the window to prevent overfitting to short subsequences. Thus, for any window $\mathbf{Y}_i$, we generate a semantically similar time series $\widetilde{\mathbf{Y}}_i = \mathbf{Y}_{i,t_0:W}$ to serve both as a pair for estimating sample-specific information and as a cross-reconstruction target. By requiring $\mathbf{\Theta}_i$ to support accurate reconstruction across varying initial conditions, the encoder learns to recover only system information while remaining invariant to information about the initial condition. This leads to the PULSE pretraining objective,

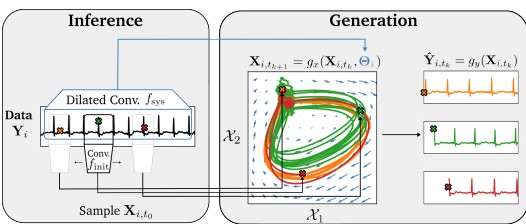

Figure 3: PULSE aims to recover system information through an inference process that uses two encoders, $f_{\text{sys}}$ to estimate shared parameters of a latent dynamical systems and $f_{\text{init}}$ to estimate sample-specific initial conditions. By requiring $\mathbf{\Theta}_i$ to support reconstruction of randomly sampled $\mathbf{X}_{i,t_0}$, we encourage the recovered system information to be invariant to the sample-specific ones.

$$\mathcal{L}_{\text{PULSE}}(\mathbf{Y}_i) = \mathbb{E}_{t_0} \left[ \sum_{k=1}^{W} \|\widetilde{\mathbf{Y}}_{i,t_k} - g_y(g_x(\mathbf{X}_{i,t_{k-1}}, \mathbf{\Theta}_{i,t_k})\|^2 \right], \tag{3}$$

where $\mathbf{X}_{i,t_0} = [f_{\text{init}}(\mathbf{Y}_i)]_{t_0}$ and $\mathbf{\Theta}_i = f_{\text{sys}}(\mathbf{Y}_i)$. We find that using multiple $\widetilde{\mathbf{Y}}_i$'s to estimate the expectation can improve performance, and in our experiments we use up to four samples.

**Regularizing Time-Varying System Variables.** Since both encoders observe the same $Y_i$, and the system representation includes $\theta_{i,tk}$ derived from that same input, it is possible that the encoder could learn a trivial representation of the dynamics by simply copying local signal values into these $\theta_{i,tk}$. We employ the following two strategies to mitigate this scenario, both based on limiting the expressivity of $\theta_{i,tk}$. First, we constrain the dimensionality of the $\theta_{i,tk}$ to a single dimension similar to prior works (Li and Mandt, 2018). This ensures that $\theta_{i,tk}$ alone does not have sufficient capacity to represent the full diversity of initial conditions in the data, particularly for multi-channel observations. Second, we limit the temporal variability of $\theta_{i,k}$ by reducing its ability to change between timesteps. Specifically, we apply adaptive max pooling across the time dimension for $\theta_{i,tk}$ and assign the pooled values back to their corresponding time steps, effectively sharing each pooled component across multiple consecutive timesteps.

### 3.3 PROVABLE RECOVERY OF SYSTEM INFORMATION

We now provide a theoretical analysis of our framework and identify conditions under which cross-reconstruction provably recovers system information. Our strategy is to build on prior work showing that MAE pretraining implicitly recovers information from the minimal set of latent variables shared[1] $\mathcal{C}$ between masked and unmasked regions in a hierarchical data-generating process (Kong and Zhang, 2023). By viewing cross-reconstruction as an MAE task under a specific masking strategy, we can extend this theory to characterize how different masking choices determine the type of information recovered in our own time-series generative model.

Cross-reconstruction can be viewed as an MAE task by treating the pair $(\mathbf{Y}_i, \mathbf{Y}_j)$ as a single joint input with a pair of masking variables $(\mathbf{m}_i, \mathbf{m}_j)$, where each $\mathbf{m}_i \in \{0,1\}^{W \times M}$ indicates for every element $(w, m)$ whether it is observed ($m_{i,w,m} = 1$) or masked ($m_{i,w,m} = 0$). In this view, Eq. 2 corresponds to setting $\mathbf{m}_{i,1:W,1:M} = 1$ to retain $\mathbf{Y}_i$ as input, while fully masking the other sample $\mathbf{m}_{j,1:W,1:M} = 0$, so that $\mathbf{Y}_j$ is removed and can serve as the reconstruction target. The effect of this masking strategy on the type of information recovered can then be characterized given our generative model in Eq. 1. To make this precise, we outline assumptions on the generative process.

**Assumption 1** (Data-Generating Process). *The process in Equation* (1) *satisfies the following conditions: (i) the fully factorized generative model is a directed acyclic graph (DAG); and (ii) each function $g_k$ is invertible.*

---

[1]Our minimal set of shared latent variables $\mathcal{C}$ is defined in Theorem 1 of (Kong and Zhang, 2023) as $\mathbf{c}$.

**Assumption Interpretation.** Part (i) ensures that our theory applies to complex systems with elaborate parameter factorizations, as long as they remain acyclic. Part (ii) guarantees that no information is lost during the generative process and is adopted from prior work on identifiable deep generative models (Locatello et al., 2020; Von Kügelgen et al., 2021).

Given this data-generating process and the cross-reconstruction masking scheme described above, we present our theory, which identify the $\mathcal{C}$ between masked and unmasked regions. This $\mathcal{C}$ corresponds to the information that is implicitly recovered during MAE pretraining (Kong and Zhang, 2023).

**Theorem 1.** *Given two time series $\mathbf{Y}_i$ and $\mathbf{Y}_j$ independently sampled from the same system (i.e., $\boldsymbol{\Theta}_i = \boldsymbol{\Theta}_j = \boldsymbol{\Theta}^{(s)}$) under the generative process defined by Eq. 1 and Assumption 1, the minimal set of latent variables shared is the system parameters $\boldsymbol{\Theta}^{(s)}$ if and only if all observables from one series is fully masked (i.e., $\mathbf{m}_{i,1:W,1:M} = 0$ and $\mathbf{m}_{j,1:W,1:M} = 1$).*

A proof is provided in Appendix A. Theorem 1 states that system information is recovered during a masked reconstruction task when an entire time series is removed from the input pair, as this masking scheme uniquely ensures that the $\mathcal{C}$ connecting the masked and unmasked regions contains the system parameters. Importantly, this reconstruction task with whole-sample masking corresponds exactly to the pretraining objective $\mathcal{L}_{\text{Cross}}$. Moreover, this theory predicts that when a time series contains both masked and unmasked regions, $\mathcal{C}$ necessarily includes the state variables $\mathbf{X}$, causing the recovered information to confound sample-specific and system information.

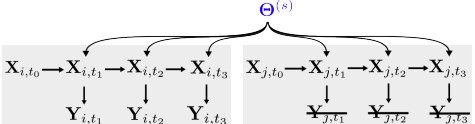 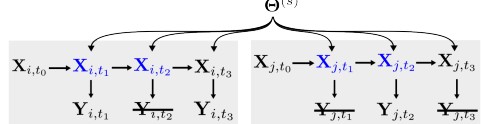

A) Positive Mask: System Information Recovered    B) Negative Mask: Sample-Specific Information Recovered

Figure 4: We illustrate how these different masking strategies recover different sources of information in our data-generating process for sample pairs $(\mathbf{Y}_i, \mathbf{Y}_j)$ where $i, j \in \mathcal{I}_s$. $\cancel{\mathbf{Y}}$ marks an observable that is removed from the input and used as a reconstruction target. **Blue** highlights $\mathcal{C}$, representing the information that is recovered during pretraining. Theorem 1 predicts that $\mathcal{C} = \{\boldsymbol{\Theta}^{(s)}\}$ only when information from one sample is fully removed. Gray boxes group latent variables that are specific to each time-series sample.

Since $\mathcal{L}_{\text{PULSE}}$ can be viewed as an approximation of $\mathcal{L}_{\text{Cross}}$ that uses pseudo-pairs to simulate independent samples, this theory offers an explanation for how $\mathcal{L}_{\text{PULSE}}$ recovers system information. A discussion on how PULSE approximates the cross-reconstruction process for independent samples is provided in Appendix I.

## 4 SYNTHETIC DYNAMICAL SYSTEMS EXPERIMENTS

**Set up.** We investigate how our findings in Theorem 1 extrapolate to settings where Assumption 1 may not completely hold. Specifically, we consider a synthetic dynamical systems experiment, where $g_x$ can not be practically inverted due to system chaos. The goal of this task is to learn a representation space that can distinguish between parameter settings while remaining robust to increasing levels of dynamical system noise. This task is motivated by real-world scenarios in which parameter changes in the underlying system may correspond to shifts in physiological state, and effective representations must reliably capture these shifts. Time series data are generated from three stochastic differential equations (Lorenz, Thomas, and Hindmarsh-Rose) across a grid of parameters in bifurcation regions and noise levels $\sigma = \{0, 1, 3, 5\}$. Datasets are then constructed by randomly selecting five parameter settings to form a five-class classification problem, with trials split into 70:15:15 for train, validation, and test sets and subsequently segmented into windows where $W = 100$. For each system at each noise level, we average classification accuracy over 10 random seeds over both dataset sampling and model initialization. The reported values are averaged across all three systems. Full dataset details are provided in Appendix B.

**Results.** Table 1 shows that PULSE consistently achieves the highest classification accuracy among all practical baselines (described in Section 5 and Appendix C), even as $\sigma$ increases. This demonstrates that PULSE pretraining on pseudo-pairs $(\mathbf{Y}_i, \widetilde{\mathbf{Y}}_i)$ is more robust to dynamical system noise

and can extract class-discriminative features that reveal changes in system parameters under noisy conditions. Furthermore, we validate Theorem 1 by considering a PULSE Oracle where label information is used to identify true $(\mathbf{Y}_i, \mathbf{Y}_j)$ pairs to construct the positive and negative set ups illustrated in Figure 4. Specifically, the positive model leverages labels to select pairs in $\mathcal{L}_{\text{Cross}}$, whereas the negative model applies random temporal masking to each pair before inputting both into $f_{\text{init}}$ and $f_{\text{sys}}$. According to Theorem 1, the negative oracle captures sample-specific information that is uninformative for classifying system parameters, so the positive oracle is expected to consistently outperform it.

| $\sigma$ | SimCLR | TS2Vec | REBAR | PatchTST | TimeMAE | LFADS | DSVAE | PULSE | PULSE Oracle Positive | Negative |
|---|---|---|---|---|---|---|---|---|---|---|
| 0 | 93.08 | 98.68 | 98.90 | 77.59 | 96.29 | 99.06 | 98.56 | **99.58** | 99.29 | 98.86 |
| 1 | 83.10 | 93.07 | 93.36 | 50.36 | 93.42 | 93.02 | 90.84 | **96.09** | **97.26** | 96.66 |
| 3 | 70.05 | 79.78 | 79.36 | 39.88 | 75.08 | 79.03 | 76.70 | **83.42** | **89.00** | 84.62 |
| 5 | 62.29 | 73.67 | 72.37 | 37.82 | 66.63 | 71.33 | 69.90 | **77.34** | **82.65** | 76.90 |

Table 1: Our results confirm Theorem 1's predictions: The positive oracle improves classification accuracy over PULSE pretraining without labels, while the negative oracle reduces performance relative to the positive model and, at $\sigma = 5$, performance even falls below the practical PULSE algorithm. Moreover, PULSE is the most effective practical algorithm, being the best at distinguishing parameter changes across all $\sigma$. **Black bold** indicates the best practical algorithm (pretrained without labels), while **blue bold** indicates the best oracle (pretrained with labels) when it exceeds practical methods. Fig. 4 illustrates the masking strategy used in the positive and negative oracle.

## 5 REAL PHYSIOLOGICAL DATA EXPERIMENTS

In this section, we describe our experimental set up for comparing PULSE pretraining with other SSL pretraining approaches across several physiological datasets and downstream tasks.

**Data.** We consider 4 commonly used physiological time-series datasets from distinct sensor domains, each consisting of long trials with time-varying classification labels. We use Human Activity Recognition (HAR) (Reyes-Ortiz et al., 2015), where human activity is estimated from accelerometer and gyroscope signals; PPG (Schmidt et al., 2018), where optical blood volume signals are used to estimate stress levels; ECG (Moody, 1983), where the heart's electrical activity is used to detect rhythm abnormalities; and EEG (Kemp et al., 2000), where the brain's electrical activity is used to estimate sleep stages. A detailed description of these datasets is provided in Appendix D.

**Baselines.** We benchmark performance against a representative set of SSL pretraining approaches, including three CL methods: SimCLR (Chen et al., 2020), TS2Vec (Yue et al., 2022), and REBAR (Xu et al., 2023), two SVAE models: LFADS (Sedler and Pandarinath, 2023; Sussillo et al., 2016) and DSVAE (Yingzhen and Mandt, 2018), and two masked modeling approaches: TimeMAE (Cheng et al., 2023) and PatchTST (Nie et al., 2022). To evaluate the pretraining objective, we use the same dilated convolution encoder architecture (Yue et al., 2022) across all CL and SVAE experiments, ensuring that performance differences reflect the quality of the pretraining rather than differences in architectures. For MAE baselines, we retain the original transformer encoders, since the architecture is often a critical component of the method and changing it can reduce performance. Additional details on baseline design are provided in Appendix C.

### 5.1 LINEAR PROBE EVALUATION

To assess the ability of pretraining to learn class-discriminative features, we train a linear probe (logistic regression) on the frozen embeddings from each pretrained model to predict the ground truth physiological class labels from each dataset. To measure the performance, we report Accuracy, AUROC, and AUPRC averaged over 5 random model initializations over a single pre-defined training-val-test split (details in Appendix D). Table 2 shows that PULSE pretraining achieves strong linear probe performance across all four datasets, performing competitively on HAR and achieving the highest overall scores on PPG, ECG, and EEG. Notably, PULSE substantially outperforms LFADS and DSVAE on ECG and PPG, highlighting that explicitly removing noise provides an clear advantage over standard VAE objectives that do not distinguish between noise and signal. Moreover, the improvements over SimCLR, TS2Vec, and REBAR show that CL's sensitivity to false positive pairs

| | Metric | SimCLR | TS2Vec | REBAR | PatchTST | TimeMAE | LFADS | DSVAE | PULSE |
|---|---|---|---|---|---|---|---|---|---|
| HAR | Accuracy ↑ | 94.65 | 93.24 | **95.35** | 83.04 | 92.25 | 93.55 | 93.55 | 93.27 |
| | AUROC ↑ | 99.38 | 99.31 | **99.65** | 97.44 | 99.14 | 99.49 | 99.36 | 99.42 |
| | AUPRC ↑ | 97.63 | 97.66 | **98.91** | 89.90 | 97.05 | 98.29 | 97.69 | 98.10 |
| PPG | Accuracy ↑ | 34.48 | 40.23 | 41.38 | 59.78 | 61.35 | 52.81 | 58.65 | **64.27** |
| | AUROC ↑ | 61.19 | 64.28 | 69.77 | 71.08 | 78.08 | 71.10 | 76.78 | **80.29** |
| | AUPRC ↑ | 36.08 | 39.59 | 44.57 | 52.91 | 56.74 | 49.59 | 55.38 | **59.89** |
| ECG | Accuracy ↑ | 69.92 | 76.12 | 81.54 | 64.40 | 69.80 | 61.84 | 70.42 | **87.41** |
| | AUROC ↑ | 82.54 | 86.56 | 91.46 | 70.96 | 76.61 | 71.69 | 82.88 | **94.93** |
| | AUPRC ↑ | 80.63 | 85.16 | 89.85 | 68.16 | 76.62 | 69.21 | 81.31 | **94.75** |
| EEG | Accuracy ↑ | 66.38 | 83.76 | 83.71 | 80.62 | 83.83 | 82.43 | 84.25 | **85.56** |
| | AUROC ↑ | 85.45 | 94.99 | 95.08 | 93.55 | 95.09 | 94.49 | 95.42 | **96.17** |
| | AUPRC ↑ | 50.95 | 70.22 | 70.77 | 67.37 | 73.22 | 68.55 | 72.25 | **73.82** |

Table 2: Linear Probe Classification Results. PULSE performs competitively on HAR and achieves the best results on PPG, ECG, and EEG. Note that while HAR scores are lower than SOTA in this experiment, this representation leads to improved performance in Tables 3 and 4.

| Dataset | | Supervised | SimCLR | TS2Vec | REBAR | PatchTST | TimeMAE | LFADS | DSVAE | PULSE |
|---|---|---|---|---|---|---|---|---|---|---|
| 1 % | HAR | 78.39 | 71.74 | 80.57 | 81.10 | 33.27 | 80.79 | 80.97 | 79.94 | **84.74** |
| | ECG | 45.46 | 63.83 | 62.77 | 67.56 | 57.68 | 65.15 | 57.34 | 67.60 | **84.77** |
| | PPG | 34.38 | 30.34 | 31.98 | 32.33 | 41.74 | 40.45 | 40.22 | 41.28 | **42.97** |
| | EEG | 77.76 | 56.72 | 77.39 | 77.19 | 63.45 | 70.55 | 74.19 | 78.40 | **80.69** |
| 5 % | HAR | 92.34 | 85.01 | 91.76 | 91.04 | 54.79 | 91.55 | 91.48 | 90.72 | **93.14** |
| | ECG | 69.20 | 65.00 | 63.97 | 70.12 | 60.73 | 68.73 | 60.04 | 67.66 | **84.23** |
| | PPG | 42.47 | 30.62 | 33.13 | 38.25 | 49.48 | 49.53 | 45.35 | 51.15 | **53.39** |
| | EEG | **84.90** | 61.98 | 77.09 | 76.75 | 73.48 | 77.50 | 75.19 | 78.17 | 80.45 |

Table 3: Semi-supervised classification accuracy for 1% and 5% of labels averaged over 25 random seeds. Higher score is better. PULSE outperforms all SSL baselines and most supervised baselines.

may limit its effectiveness across diverse sensor domains. Finally, the performance increase over PatchTST and TimeMAE demonstrates that designing a generative task to explicitly extract system information produces representations that better distinguish physiological states than those learned by standard masked modeling, which does not leverage this structure.

## 5.2 SEMI SUPERVISED EVALUATION

Next, we evaluate the label efficiency of the pretrained representations using a semi-supervised classification task on the pretrained frozen embeddings. For each pretrained model, we train a linear probe on 1% and 5% of the ground truth labels and apply Laplace smoothing so that all downstream classes are represented. Reported accuracies are averaged over five random label subsets for each of the five model initializations from Section 5.1, for a total of 25 seeds. We also include a supervised baseline to estimate performance achievable without pretraining. Table 3 shows that PULSE pretraining consistently outperforms the baselines across all four sensor domains, achieving the highest scores among SSL methods across all datasets. Interestingly, PULSE's HAR representation is more label-efficient, achieving strong performance with very few labels despite lower linear probe scores on the full labeled dataset, suggesting that it captures key class-discriminative features more efficiently. PULSE also outperforms most supervised baselines, highlighting the advantage of its pretrained representations in limited data scenarios.

Furthermore, our results demonstrate that PULSE is broadly effective across datasets with very diverse signal properties. Each dataset represents a distinct physiological process, including human movement (HAR), the cardiovascular system (PPG, ECG), and brain activity (EEG), each with different characteristics such as short (2.56s, HAR) versus long (60s, PPG) context lengths, single-channel (PPG) versus multi-channel (6 channels, HAR) setups, and quasi-periodic (ECG) versus non-periodic (EEG) signals. PULSE achieves strong performance across this range of physiological processes and dataset settings, highlighting that extracting system information is a general principle for learning label-efficient representations from physiological signals.

## 5.3 TRANSFER LEARNING EVALUATION

We investigate in-domain transfer learning for classification in two scenarios from Zhang et al. (2022). This task is motivated by real-world applications where we want to transfer knowledge between datasets collected from similar sensors. In the first scenario, a model is pretrained on EEG (Kemp et al., 2000) and fine-tuned on the Epilepsy dataset (Andrzejak et al., 2001). In the second, a model is pretrained on HAR (Reyes-Ortiz et al., 2015) and fine-tuned on Gesture (Liu et al., 2009). Our setup follows Zhang et al. (2022), where we attach a 2-layer MLP head and fine-tune for 40 epochs using the

|  | EEG → Epilepsy | | HAR → Gesture | |
|---|---|---|---|---|
|  | ACC | AUROC | ACC | AUROC |
| SimCLR | 93.52 | 97.52 | 78.83 | 93.80 |
| TS2Vec | 93.95 | 95.87 | 77.67 | 95.45 |
| REBAR | 95.27 | 98.33 | 78.17 | 95.54 |
| PatchTST | 95.03 | 98.04 | 77.00 | 95.21 |
| LFADS | 94.71 | 98.01 | 78.50 | 95.42 |
| DSVAE | 94.97 | 98.17 | 78.00 | 95.34 |
| PULSE | **95.82** | **98.51** | **83.67** | **97.20** |

Table 4: In-Domain Transfer Learning.

Adam optimizer with a learning rate of 0.0003. A detailed description of the fine-tuning datasets is provided in Appendix D. Table 4 reports accuracy and AUROC averaged over five random seeds for both model initialization and fine-tuning. In both scenarios, PULSE consistently achieves the highest transfer performance, with a substantial gain on the HAR-to-Gesture task. This demonstrates that pretraining designed to explicitly prioritize system information can produce features that transfer effectively to related downstream tasks.

## 5.4 ABLATION STUDY

Table 5 shows the effect of ablating various PULSE components on the linear probe accuracy across all datasets, reported as $\Delta = \mathrm{Acc}_{\mathrm{Ablated}} - \mathrm{Acc}_{\mathrm{PULSE}}$. In *w/o TV-Params* $\tilde{\theta}_{i,t_k}$, we remove time-varying parameters and retain only time-invariant ones, i.e., $\mathbf{\Theta}_i = \theta_i$. This results in an average accuracy drop of 7.6%, underscoring the importance of modeling non-stationary dynamics in physiological time series. In *Shared* $f_{\mathrm{sys}}$ *and* $f_{\mathrm{init}}$, we estimate $\mathbf{X}_{i,t_k}$ from the output of $f_{\mathrm{sys}}$ such that

|  | HAR | ECG | PPG | EEG | Avg. |
|---|---|---|---|---|---|
| PULSE | **93.27** | **87.41** | **64.27** | **85.56** | - |
| w/o TV-Params $\tilde{\theta}_{i,t_k}$ | -1.73 | -6.83 | -9.22 | -12.7 | -7.62 |
| Shared $f_{\mathrm{sys}}$ and $f_{\mathrm{init}}$ | -1.02 | -1.75 | -1.58 | -0.54 | -1.22 |
| w/o Sample $t_0$ (Direct Recon. ) | -1.16 | -7.73 | -15.96 | -0.81 | -6.42 |
| Cross-Recon. w Random Pairs | -2.42 | -22.99 | -6.96 | -6.35 | -9.68 |

Table 5: Relative effect of ablation on linear probe accuracy. Baseline accuracy (top row) and relative effect ($\Delta$) of removing components (other rows). Avg. reports the mean $\Delta$ across datasets.

$\mathbf{X}_{i,t_k} = [f_{\mathrm{init}}(f_{\mathrm{sys}}(\mathbf{Y}_i))]_{t_k}$, rather than using two separate encoders. This results in a 1.22% drop, suggesting that explicitly separating transferrable and non-transferrable information can improve time-series representation learning. In *w/o Sample* $t_0$, we no longer construct pseudo-pairs by randomly sampling $t_0$. Instead, we fix $t_0 = 1$ for all samples during initial-condition inference. This effectively reduces the method to direct reconstruction, as the input–output pairs become fixed rather than randomly sampled. This leads to a 6.4% drop in average accuracy, showing that training the system representation with randomly sampled outputs is important and that our pseudo-pair strategy can effectively constructs similar time-series pairs without labels. In *Cross Reconstruction with Random Pairs*, we form input–output pairs by randomly sampling segments without regard for the underlying dynamics of each time-series sample. This ablation has the largest effect, decreasing the average performance by -9.68%. This emphasizes that effective cross-reconstruction requires identifying similar time-series and further highlighting the effectiveness of our pseudo-pair strategy in achieving this.

## 6 CONCLUSION

In this work, we introduced PULSE, an improved SSL pretraining framework that improves performance across diverse tasks by preserving system information and filtering out sample-specific noise. By explicitly using dynamical systems to guide the design of time-series SSL objectives, we hope to inspire future research on formulating SSL pretraining strategies based on an underlying generative model to obtain improved representations of more complex physiological phenomena.

# 7 REPRODUCIBILITY STATEMENT

Upon acceptance, we will release our GitHub code publicly. This will include code for downloading and preprocessing the datasets used in our work, training our approach, and evaluating results. It will also provide baseline implementations and the configuration files used to run our experiments as well as model checkpoints.

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

APPENDIX

## A    PROOF FOR THEOREM 1

**Theorem 1.** *Given two time series $\mathbf{Y}_i$ and $\mathbf{Y}_j$ independently sampled from the same system (i.e., $\mathbf{\Theta}_i = \mathbf{\Theta}_j = \mathbf{\Theta}^{(s)}$) under the generative process defined by Eq. 1 and Assumption 1, the minimal set of latent variables shared is the system parameters $\mathbf{\Theta}^{(s)}$ if and only if all observables from one series is fully masked (i.e., $\mathbf{m}_{i,1:W,1:M} = 0$ and $\mathbf{m}_{j,1:W,1:M} = 1$).*

*Proof.* Our proof relies on Theorem 1 from Kong and Zhang (2023), which establishes that the minimal set of shared information $\mathbf{c}$ in a hierarchical DAG is unique and can be identified using Algorithm 1 presented in their work. We describe the generative model that is considered and summarize Algorithm 1 before applying it in the proof.

**Two-sample Generative Model.** We consider the cross-reconstruction setting with a pair of time series $(\mathbf{Y}_i, \mathbf{Y}_j)$ for $i, j \in \mathcal{I}_s$, generated from the same system such that $\mathbf{\Theta}_i = \mathbf{\Theta}_j = \mathbf{\Theta}^{(s)}$. When considering only two samples, the joint distribution in Eq. 1 reduces to the following form,

$$p(\mathbf{Y}_i, \mathbf{Y}_j, \mathbf{X}_i, \mathbf{X}_j, \mathbf{\Theta}^{(s)}) = p(\mathbf{\Theta}^{(s)}) \prod_{n=\{i,j\}} p(\mathbf{X}_{n,t_0}) \left[ \prod_{k=1}^{W} p(\mathbf{Y}_{n,t_k} | \mathbf{X}_{n,t_k}) \right]$$
$$\left[ \prod_{k=2}^{W} p(\mathbf{X}_{n,t_k} | \mathbf{X}_{n,t_{k-1}}, \mathbf{\Theta}^{(s)}) \right]. \quad (4)$$

Moreover, we denote masked and unmasked observables respectively as,

$$\mathbf{Y_m} = \bigcup_{k=\{i,j\}} \{\mathbf{Y}_{k,t} \mid \mathbf{m}_{k,t} = 0\} \quad \text{and} \quad \mathbf{Y_{m_c}} = \bigcup_{k=\{i,j\}} \{\mathbf{Y}_{k,t} \mid \mathbf{m}_{k,t} = 1\}. \quad (5)$$

In words, $\mathbf{Y_m}$ and $\mathbf{Y_{m_c}}$ are the sets of masked and unmasked time-points across both samples, respectively.

**Algorithm 1 from Kong and Zhang (2023).** This approach uses a two-stage procedure for identifying the minimal set of latent variables $\mathcal{C}$ shared between masked and unmasked observables in a hierarchical graphical model. In the first stage, called the *selection stage*, all latent variables that are ancestors of both $\mathbf{Y_m}$ and $\mathbf{Y_{m_c}}$ are located. This is done by collecting all parents of the masked observables and adding those that are also ancestors of the unmasked observables to $\mathcal{C}$. The resulting set $\mathcal{C}$ contains all latent variables shared between $\mathbf{Y_m}$ and $\mathbf{Y_{m_c}}$. In the second stage, called the *pruning stage*, each element of $\mathcal{C}$ is checked to ensure that no other element in $\mathcal{C}$ lies on the directed path between it and $\mathbf{Y_{m_c}}$. If such a descendant exists, the ancestor node is removed from $\mathcal{C}$. To summarize, given a hierarchical graphical model with masked and unmasked observables $\mathbf{Y_m}$ and $\mathbf{Y_{m_c}}$, this algorithm returns $\mathcal{C}$, the set of variables with lowest dimension that block all paths between $\mathbf{Y_m}$ and $\mathbf{Y_{m_c}}$.

Now, we prove each direction of the if-and-only-if condition separately.

**If a full sample is masked, then $\mathcal{C} = \mathbf{\Theta}^{(s)}$.** Without loss of generality, we define full-sample mask as $\mathbf{m}_i = 0$ and $\mathbf{m}_j = 1$ for $i, j \in \mathcal{I}_s$. In this case, any element in $\mathcal{C}$ must be a common parent of both samples. As illustrated in Fig. 4A, the only parent node that is shared between $\mathbf{Y}_i$ and $\mathbf{Y}_j$ is $\mathbf{\Theta}^{(s)}$, and thus the only set of variables shared between masked and unmasked regions is the system variables. When Algorithm 1 from (Kong and Zhang, 2023) is applied to the graphical model in Eq. 4 under the full-sample masking scheme, it recovers $\mathcal{C} = \{\mathbf{\Theta}^{(s)}\}$. In the selection stage, $\mathbf{\Theta}^{(s)}$ is the only parent node connecting both samples $\mathbf{Y}_i$ and $\mathbf{Y}_j$. In the pruning stage, nothing is removed since $\mathcal{C}$ contains only a single element, implying no additional shared latent variables exist as children of $\mathbf{\Theta}^{(s)}$. Therefore, under full-sample masking, the minimal set of shared latent variables includes only the shared system parameters $\mathbf{\Theta}^{(s)}$.

**If $\mathbf{c} = \mathbf{\Theta}^{(s)}$, then a full sample is masked.** We prove the statement via its contrapositive: if a sample is not fully masked (i.e. contains both masked and unmasked observables), then the minimal set of shared latent variables cannot consist solely of the system parameters.

Intuitively, as shown in Fig. 4B, when a sample contains both masked and unmasked time-points, there is always a latent state variable $\mathbf{X}$ that serves as the parent node connecting $\mathbf{Y_m}$ and $\mathbf{Y_{m_c}}$. To formalize this, we define a *subsequence mask* as a consecutive region of masked observables, i.e., $\mathbf{m}_{i,t_0:t_1} = 0$, where $t_0, t_1 \in 1, \ldots, T$, $t_0 \leq t_1$, and $t_1 - t_0 < T$ for a partial mask in a sample of length $T$. Thus, $t_0 = t_1$ corresponds to masking a single time point at the index $t_0$.

There are three possible types of masked subsequence regions: (1) a mask bordering the left edge, or the beginning of the sample, (2) a mask bordering the right edge, or the end of the sample, and (3) a mask in the middle of the sample that is bordered on the left and right by unmasked regions. We apply Algorithm 1 in Kong and Zhang (2023) to each case to determine what minimal set of shared latent variables are recovered. We use $\mathcal{C}_{\text{subseq}}$ to denote the minimal set of shared latent variables between masked and unmasked regions induced by a subsequence mask.

*Case 1 (Left Edge):* When the mask borders the left edge of sample $\mathbf{Y}_i$, the masked region satisfies $t_0 = 1$ and $t_1 < T$. During the selection stage, Algorithm 1 retrieves $\mathcal{C}_{\text{subseq}} = \{\boldsymbol{\Theta}^{(s)}, \mathbf{X}_{i,t_1}\}$, since these nodes are ancestors of both masked and unmasked regions. In the pruning stage, $\boldsymbol{\Theta}^{(s)}$ is removed because $\mathbf{X}_{i,t_1}$ lies on the directed path between $\boldsymbol{\Theta}^{(s)}$ and the unmasked region. Therefore, $\mathcal{C}_{\text{subseq}} = \{\mathbf{X}_{i,t_1}\}$. Intuitively, the latent variable $\mathbf{X}_{i,t_1}$ serves as the minimal parent connecting the masked and unmasked unmasked regions, $\mathbf{Y}_{i,1:t_1}$ and $\mathbf{Y}_{i,t_1:T}$ respectively.

*Case 2 (Right Edge):* The right edge follows a similar analysis, where the masked region satisfies $t_0 > 1$ and $t_1 = T$. This results in $\mathcal{C}_{\text{subseq}} = \{\boldsymbol{\Theta}^{(s)}, \mathbf{X}_{i,t_0}\}$ after the selection stage, and $\mathcal{C}_{\text{subseq}} = \{\mathbf{X}_{i,t_0-1}\}$ after the pruning stage. Thus, the latent variable above the $\mathbf{X}_{i,t_0-1}$ serves as the lowest-level parent connecting unmasked and masked regions, $\mathbf{Y}_{i,1:t_0}$ and $\mathbf{Y}_{i,t_0:T}$ respectively.

*Case 3 (Middle):* When $t_0 > 1$ and $t_1 < T$, the masked region is bordered on both the left and right by unmasked variables. During the selection stage, $\mathcal{C}_{\text{subseq}} = \{\mathbf{X}_{i,t_0-1}, \mathbf{X}_{i,t_1}, \boldsymbol{\Theta}^{(s)}\}$, and after pruning, $\mathcal{C}_{\text{subseq}} = \{\mathbf{X}_{i,t_0-1}, \mathbf{X}_{i,t_1}\}$, since the latent state variables lie on the directed paths from the system variables to the unmasked regions. Therefore, there are two minimal parent nodes: one at the left boundary, $\mathbf{X}_{i,t_0-1}$, above the unmasked variable $\mathbf{Y}_{i,t_0-1}$, and one at the right boundary, $\mathbf{X}_{i,t_1}$, above the masked variable $\mathbf{Y}_{i,t_1}$.

Putting these results together, the minimal shared latent variables induced by a subsequence mask is given by,

$$
\mathcal{C}_{\text{subseq}} = \begin{cases} \{\mathbf{X}_{i,t_1}\}, & \text{if } t_0 = 1 \text{ and } t_1 < T, \\ \{\mathbf{X}_{i,t_0-1}\}, & \text{if } t_0 > 1 \text{ and } t_1 = T, \\ \{\mathbf{X}_{i,t_0-1}, \mathbf{X}_{i,t_1}\}, & \text{if } t_0 > 1 \text{ and } t_1 < T. \end{cases} \tag{6}
$$

We can extend this result to arbitrary masks, since any masking configuration over timepoints can be expressed as a union of subsequence masks, $\mathbf{m}_i = \bigcup_k \mathbf{m}_{i,t_0^{(k)}:t_1^{(k)}}$. To enforce partial masking, we require that $\mathbf{m}_{i,t} = 0$ for some $t$ and $\mathbf{m}_{i,t'} = 1$ for some $t' \neq t$. Consequently, for arbitrary masks, the minimal shared latent set is the union over the minimal sets for each subsequence mask $\mathcal{C} = \bigcup_k \mathcal{C}_{\text{subseq}}^{(k)}$. Since each $\mathcal{C}_{\text{subseq}}$ contains only latent state variables, the union $\mathcal{C}$ does not include the system variables. Therefore, we have shown that under partial masking, the minimal set of shared latent variables does not include the system parameters.

To summarize, applying Algorithm 1 from Kong and Zhang (2023) under a full-sample masking scheme yields $\mathcal{C} = \{\boldsymbol{\Theta}^{(s)}\}$, since it is the only variable shared between samples in our graphical model. In contrast, under partial masking, the minimal shared latent set includes only the latent state variables at the boundaries of the masked subsequences, and never the system parameters. This follows directly from our hierarchical structure, as the $\mathbf{X}$ variables always lie on the directed path from $\boldsymbol{\Theta}^{(s)}$ to $\mathbf{Y}_{\mathbf{m_c}}$.

Thus, under our two-sample generative model and applying Algorithm 1 from Kong and Zhang (2023), the system parameters appear as the minimal shared latent variables if and only if an entire sample is masked.

$\square$

## B  SYNTHETIC DATASET DESCRIPTION

Synthetic time-series trials $\mathbf{y} \in \mathbb{R}^{T \times M}$ of length $T$ and $M$ measurement dimensions are generated by numerically integrating the Stratonovich SDE,

$$d\mathbf{y}_t = f(\mathbf{y}_t)dt + \tilde{\sigma}d\mathbf{B}_t, \tag{7}$$

with a fixed step size $dt = 10^{-3}$, where $f(\cdot)$ is the dynamics function, $\mathbf{B}_t$ is multidimensional Brownian noise. To ensure that the noise levels are comparable across different systems, we set the diffusion scale as $\tilde{\sigma} = \sigma \, \mathrm{RMS}(\mathbf{Y})$ where $\sigma$ is a dimensionless noise level and $\mathrm{RMS}(\mathbf{Y})$ is the root-mean-square amplitude of the noiseless time-series and is estimated empirically through samples $\mathbf{Y}$ from the system. We consider the following noise levels $\sigma = \{0, 1, 3, 5\}$ and integrate eq. 7 using `torchsde` (Li et al., 2020; Kidger et al., 2021).

We generate time series from three dynamical systems: Lorenz, Thomas, and Hindmarsh-Rose. These define strange attractor that produces bounded yet nontrivial dynamics. Importantly, for certain parameter regimes, these systems undergo bifurcations, where changes to parameters induce qualitative changes in the dynamics, thereby altering the statistical properties of the resulting time series. We select systems whose behavior is sensitive to parameter changes, as physiological time series are also generated by nonlinear dynamical systems that are highly sensitive to changes in their underlying parameters. For example, the bursting behavior of a neuron can be triggered or suppressed depending on the inputs it receives (Hindmarsh and Rose, 1984; Kim and Lim, 2019). For each parameter setting and noise level, we generate 20 long time series with $T = 10^5$ time-steps from random initial conditions $\mathbf{Y}_{i,0} \sim \mathcal{N}(0, I)$ and discard the first 200 steps as a burn-in period to ensure convergence to the attractor manifold. Below, we detail the parameters that we consider for each system.

**Lorenz (Lorenz, 1963).** Although originally derived from atmospheric convection, the Lorenz attractor serves as a conceptual tool for studying physiological dynamics. It exhibits bounded, irregular, and parameter-sensitive behavior, features shared by many biological systems such as heart rate variability (Billman, 2011) and neural activity (Chen et al., 2024; Mudrik et al., 2024; Sussillo et al., 2016). This is the canonical 3D nonlinear attractor used to study chaotic behavior in dynamical systems, with a state-space trajectory that resembles butterfly wings. For this system, $M = 3$, and the dynamics are given by,

$$\frac{d\mathbf{y}}{dt} = \begin{bmatrix} s(y_2 - y_1) \\ y_1(\rho - y_3) - y_2 \\ y_1 y_2 - \beta y_3 \end{bmatrix} \tag{8}$$

where $\mathbf{y} = [y_1, y_2, y_3]^\top$. Following prior work (Sparrow, 2012; Kamiya et al., 2024), we fix $\beta = 8/3$ and $s = 28$, and sweep $\rho$ across the following 10 values: $\{28, 41, 55, 69, 83, 96, 110, 124, 138, 152\}$. These values span a range of distinct chaotic regimes.

**Thomas (Thomas, 1999).** The Thomas attractor is a 3D strange attractor that produces cylindrically symmetric time series in state space. For this system, $M = 3$, and the dynamics are given by

$$\frac{d\mathbf{y}}{dt} = \begin{bmatrix} \sin(y_2) - by_1 \\ \sin(y_3) - by_2 \\ \sin(y_1) - by_3 \end{bmatrix} \tag{9}$$

where we sweep over $b \in \{0.025, 0.05, 0.075, 0.1, 0.125, 0.15, 0.175, 0.2, 0.225, 0.25\}$, corresponding to 10 equally spaced values from 0.025 to 0.25 in increments of 0.025. This range was chosen based on prior work (Sorin and Tulchinsky, 2024), which demonstrates significant changes in system behavior between values of 0 and 0.33.

**Hindmarsh-Rose (Hindmarsh and Rose, 1984).** This is a 3D dynamical systems model of neuronal activity that exhibits bursting behavior. For this system, $M = 3$, and the dynamics are given by,

$$\frac{d\mathbf{y}}{dt} = \begin{bmatrix} y_2 - ay_1^3 + by_1^2 - y_3 + I \\ c - dy_1^2 - y_2 \\ r[s(y_1 - x_R) - y_3] \end{bmatrix}, \tag{10}$$

and we sweep the external current parameter $I = \{1, 1.33, 1.66, 2, 2.33, 2.66, 3, 3.33, 3.66, 4\}$, corresponding to 9 equally spaced values from 1 to 4 in increments of 0.33. This range of parameters is chosen based on prior work (Chen et al., 2023a; Dhamala et al., 2004; Goufo et al., 2020), which shows that the system exhibits different spike–burst behaviors within this region.

Given these generated time-series trials, we construct a dataset for each system, parameter setting, and noise level by combining data from five randomly selected parameter values from each system's grid. By randomly selecting these parameter values, we ensure a range of task difficulty where more challenging datasets involve classifying parameter values that are close together, whereas easier datasets involve classifying parameter values that are farther apart. Importantly, we split each trial into 70:15:15 train, validation, and test splits, and then segment each trial into non-overlapping windows of size $W = 100$. For each system and noise level, we measure the classification accuracy averaged over ten random seeds, accounting for both dataset sampling (classification difficulty) and model initialization. The results reported in Table 1 are the average result for all three systems.

## C  BASELINE DESCRIPTION

For contrastive learning and SVAE baselines, we fix the encoder across different training objectives to control for the effects of encoder design and isolate the impact of the pretraining objective. Specifically, we adopt the time series encoder from TS2Vec (Yue et al., 2022), which consists of a 10-layer dilated convolutional network with an embedding size of 320. This setup follows the experimental protocol of (Xu et al., 2023) and report the best available performance from prior work or our own experiments. To obtain a representative embedding for each time window, we apply a global max pooling layer to aggregate features across the temporal dimension. Below, we describe each baseline in more detail.

**SimCLR (Chen et al., 2020).** SimCLR is a simple augmentation-based method that we adapt for time series data to evaluate the effectiveness of a purely augmentation-driven strategy. Three standard augmentations are applied, each with a 50% probability: scaling, which multiplies the entire time series by a factor drawn from $U(0.5, 1.5)$; shifting, which offsets the time series by a random value in the range $[-\text{subsequence size}, \text{subsequence size}]$; and jittering, which adds Gaussian noise with a standard deviation equal to 0.2 times the standard deviation of the dataset.

**TS2Vec (Yue et al., 2022).** TS2Vec is a competitive time-series contrastive learning method for time series that learns time-stamp-level representations through augmentations. It employs hierarchical contrast, combining instance-level and temporal contrast across multiple resolutions to capture scale-invariant representations within augmented context views. In our experiments, we adopt the dilated convolutional encoder from this method and use the official implementation available at `https://github.com/yuezhihan/ts2vec`.

**REBAR (Xu et al., 2023).** REBAR is a recent time-series contrastive learning method that defines positive pairs using a learned similarity measure. This is accomplished through a cross-attention mechanism that identifies class-specific motifs in one subsequence that can be used to reconstruct another. Subsequences that have the lowest reconstruction error are selected as positive pairs for contrastive learning. We use the implementation provided in the official repository: `https://github.com/maxxu05/rebar`.

**PatchTST (Nie et al., 2022).** PatchTST is a transformer model developed for forecasting that uses a patching mechanism, in which consecutive blocks of time points are processed together, and incorporates channel independence, processing each channel separately. It achieves strong performance in forecasting tasks and has been used as a backbone for extracting information about underlying physiological states from biosignals (Geenjaar and Lu, 2025). While the original paper focuses primarily on supervised learning, the model can also be trained in a self-supervised fashion using a masked autoencoding (MAE) objective. In our experiments, we use the Hugging Face implementation from the official repository `https://github.com/yuqinie98/PatchTST` and train PatchTST in SSL mode, using the CLS token as the summary representation for downstream evaluations.

**TimeMAE (Cheng et al., 2023).** TimeMAE explores the idea of block masking in a MAE framework, adapting it to time-series data. During training, random segments of the input time series are masked, and the unmasked portions are passed through an encoder to produce latent repre-

sentations. The model also introduces a decoupled autoencoder, where masked and unmasked regions are encoded separately, allowing it to extract transferable information between these regions. A lightweight decoder then reconstructs the masked segments, and the model is trained to minimize the reconstruction error. We use the implementation from the official repository: `https://github.com/Mingyue-Cheng/TimeMAE`.

**LFADS (Sussillo et al., 2016; Sedler and Pandarinath, 2023).** Latent Factor Analysis via Dynamical Systems (LFADS) is a deep generative model designed to uncover low-dimensional latent dynamics underlying neural population activity (Pandarinath et al., 2018). During inference, a bidirectional GRU produces an initial condition that serves as a summary representation of a time-series window. This initial condition is then evolved forward by a GRU with a global dynamics function to generate a latent time series, which is linearly projected back into data space to reconstruct the original time series. Although LFADS was originally developed for neural spiking data, its Poisson likelihood can be replaced with a Gaussian likelihood to adapt the framework for continuous-valued time series. In our experiments, we modify the official codebase (`https://github.com/arsedler9/lfads-torch`) by replacing the BiGRU encoder with the same dilated convolution architecture used in previous baselines, and we find that this modification improves performance on downstream tasks.

**DSVAE (Li and Mandt, 2018).** Disentangled Sequential Variational Autoencoder (DSVAE) (Li and Mandt, 2018) is a generative model originally developed for sequential data (video and audio) and has not previously been applied to physiological signals. We include this baseline in our work since it's generative process resembles PULSE and our results show it is a competitive baseline when applied onto physiological time-series. DSVAE uses a BiLSTM and MLP to infer static and dynamic latent variables, which are then used as initial conditions and inputs to an LSTM for generation. Importantly, DSVAE does not remove irrelevant information, as both system and sample-specific latents are observed and reconstructed jointly. In contrast, PULSE explicitly discards information about noise through the its objective that leverages pseudo-pairs. We use the official implementation[2], replacing the encoder with dilated convolutions, which improves downstream performance.

**PULSE.** This is our cross-reconstruction based method for physiological time-series SSL proposed in this paper.

## D REAL DATASET DESCRIPTION

For the linear probe experiments, we partition the data into training, validation, and test sets with a 70/15/15 inter-subject splits. Note that the total time of the labeled subsequences may not match the full length of the original recordings, as some portions of the data can be unlabeled. Extracted subsequences are non-overlapping.

**HAR (Reyes-Ortiz et al., 2015).** The HAR dataset consists of time series data recorded from 30 volunteers of ages 19-48 years with a smartphone (Samsung Galaxy S II) attached at the subjects waist. There are 59 samples of 5-minute long time series that are collected at 50 hz. In our experiment, we use as input 6-channels ( 3-axis linear accelerometer and 3-axis angular velocity). 6-channels recordings Raw accelerometer and gyroscopic sensor data. Subsequences are 2.56 seconds long (128 time steps) which matches the labels from the original work. collected from smartphones. 6-class classification task with 4,600 subsequences with the following activity class label names and proportions: walking (17.7 %), walking upstairs (7.6 %), walking downstairs (9.1 %), sitting (18.2 %), standing (20.1 %), and laying (20.1%)

**ECG (Moody, 1983).** We use data from the MIT-BIH Atrial Fibrillation dataset (Moody, 1983). Since no subsequence length was defined in the original work, we adopt 10-second segments, following both prior analyses of this dataset (Tonekaboni et al., 2021; Xu et al., 2023) and the convention used in ECG classification studies more broadly (Wagner et al., 2020). In total, 76,590 distinct subsequences are extracted from 23 recordings, each lasting approximately 9.25 hours and sampled at 250 Hz with two channels. To further improve computational efficiency, each subsequence is downsampled by a factor of five and produces subsequences with 500 time-steps. Of these, 76,567 subsequences are labeled, with 41.7% corresponding to atrial fibrillation and 58.3% to normal rhythm.

---

[2]`https://github.com/yatindandi/Disentangled-Sequential-Autoencoder`

**PPG (Schmidt et al., 2018).** This dataset is constructed from the WESAD dataset. There are 15 recordings each of approximately 87 minutes in duration, corresponding to 334,080 samples collected at 64 Hz from a single channel. From these recordings we extract 1,305 distinct 1-minute subsequences, consistent with the segmentation strategy used in the original work. We improve computational efficiency by downsampling each subsequence by a factor of four, so that each subsequence has 960 time steps. Of these, 666 subsequences are annotated with class labels: baseline (42.7%), stress (24.0%), amusement (12.4%), and meditation (20.9%). All signals are denoised following the procedure described in Heo et al. (2021).

**EEG (Kemp et al., 2000).** Sleep-EDF (Kemp et al., 2000) contains 39 whole-night electroencephalography (EEG) recordings collected using sleep cassettes from 20 healthy subjects. Following the preprocessing protocol of (Chambon et al., 2018), we use two EEG leads (Fpz-Cz and Pz-Oz) to evaluate pretraining, sampled at 100 Hz and segmented into non-overlapping 30-second intervals (3,000 time steps). This yields a total of 35,424 samples with the following class distribution: Wake (22.9%), N1 (8.89%), N2 (51.30%), N3 (9.92%), and REM (7.00%). The data preprocessing is accomplished with the MNE package (Gramfort et al., 2013). For transfer learning, we follow prior experimental conditions and consider a pretrained model that uses a single EEG lead (Fpz-Cz), segmented into windows of length 200.

The following data are publicly available through the links provided in the repository at `https://github.com/mims-harvard/TFC-pretraining`.

**Epilepsy. (Andrzejak et al., 2001)** The Epilepsy dataset contains 500 single-channel EEG recordings, each lasting 23.6 seconds. To minimize subject-specific bias, the recordings are divided into 11,500 one-second segments and randomly shuffled, with signals sampled at 178 Hz. The dataset has five labels that capture different conditions or recording locations: eyes open, eyes closed, EEG from healthy regions, EEG from tumor regions, and seizure activity. For our experiments, we reduce the task to binary classification by grouping the first four categories as the negative class and using seizure episodes as the positive class. Fine-tuning is performed on a small labeled subset of 60 samples (30 per class), with 20 additional samples (10 per class) used for validation. The model that achieves the best validation performance is then evaluated on the remaining 11,420 test samples.

**Gesture. (Liu et al., 2009)** The Gesture dataset captures eight distinct hand movements recorded via accelerometers, with each gesture defined by the path of motion. The gestures include swiping left, right, up, or down; waving in clockwise or counterclockwise circles; tracing a square; and drawing a right arrow. Each gesture is assigned a unique classification label. While the original study reports 4,480 recordings, only 440 samples were available from the UCR Database, yielding a balanced dataset with 55 samples per class. The sampling frequency is not specified in the original paper but is assumed to be 100 Hz. Despite its modest size, the dataset provides enough samples for fine-tuning experiments.

# E    DECLARATION OF LLM USAGE

We used LLMs solely for polishing and revising a base version of the text, for example by converting bullet points into prose, shortening sentences, or improving clarity. LLMs were not used in any other stage of this work, such as information retrieval and discovery, research ideation, or narrative development.

# F    CROSS VALIDATION EXPERIMENTS

Here, we further validate our findings with additional cross validation experiments that assess generalization across within-dataset variation. Specifically, we perform 5-fold cross validation, with each fold trained using a different random initialization seed. Note that we also include SimMTM (Dong et al., 2023), a recent competitive contrastive MAE hybrid model, as a baseline. Table E shows that PULSE achieves the best performance across all datasets, with results consistent with the single-split scores reported in Table 2. Notably, PULSE becomes the top performer on HAR when averaged over cross validation splits, even though it was not in the single-split setting. These results indicate that

| | Metric | SimCLR | TS2Vec | REBAR | PatchTST | TimeMAE | LFADS | DSVAE | SimMTM | PULSE |
|---|---|---|---|---|---|---|---|---|---|---|
| HAR | Accuracy ↑ | 89.47 (3.47) | 92.56 (2.96) | 94.13 (2.06) | 80.48 (1.68) | 94.16 (1.37) | 93.27 (2.44) | 92.00 (3.34) | 95.19 (1.81) | **95.30 (1.72)** |
| | AUROC ↑ | 98.31 (1.23) | 99.34 (0.36) | 99.35 (0.34) | 97.30 (0.35) | 99.59 (0.11) | 99.42 (0.33) | 99.27 (0.37) | **99.73 (0.14)** | 99.73 (0.15) |
| | AUPRC ↑ | 93.69 (3.99) | 97.51 (1.31) | 97.56 (1.09) | 89.21 (1.44) | 98.37 (0.54) | 97.83 (1.19) | 97.06 (1.45) | 98.24 (0.66) | **98.99 (0.58)** |
| PPG | Accuracy ↑ | 48.03 (4.16) | 57.58 (2.68) | 60.67 (3.08) | 57.71 (3.12) | 61.92 (3.64) | 55.34 (3.31) | 60.96 (2.43) | 53.93 (4.42) | **63.87 (2.50)** |
| | AUROC ↑ | 65.22 (3.06) | 76.39 (2.15) | 78.04 (2.73) | 70.93 (1.44) | 77.61 (0.86) | 73.48 (1.79) | 77.91 (3.17) | 75.74 (2.70) | **79.77 (2.77)** |
| | AUPRC ↑ | 41.89 (2.09) | 53.02 (3.91) | 57.94 (3.77) | 53.06 (1.94) | 58.27 (1.15) | 50.52 (3.40) | 57.07 (3.56) | 51.79 (3.75) | **58.43 (2.69)** |
| ECG | Accuracy ↑ | 73.14 (7.13) | 77.76 (4.96) | 80.00 (2.53) | 68.02 (9.18) | 67.96 (11.65) | 64.60 (3.89) | 77.93 (3.15) | 82.99 (1.72) | **88.01 (1.12)** |
| | AUROC ↑ | 75.49 (8.30) | 85.96 (7.36) | 87.19 (2.19) | 72.92 (10.98) | 87.84 (5.40) | 73.60 (8.27) | 83.30 (7.70) | 94.35 (2.31) | **96.95 (0.73)** |
| | AUPRC ↑ | 71.45 (7.31) | 80.03 (10.30) | 82.50 (0.70) | 68.99 (6.43) | 84.16 (7.81) | 67.66 (7.03) | 79.13 (9.09) | 90.49 (1.22) | **95.32 (0.29)** |
| EEG | Accuracy ↑ | 69.08 (2.99) | 82.15 (1.81) | 82.27 (1.74) | 80.83 (0.36) | 80.07 (0.59) | 82.37 (1.32) | 82.74 (1.75) | 80.53 (1.82) | **84.25 (1.63)** |
| | AUROC ↑ | 88.01 (1.65) | 95.01 (0.80) | 95.05 (0.73) | 94.99 (0.15) | 94.30 (0.52) | 95.00 (0.62) | 95.17 (0.60) | 94.65 (0.74) | **96.33 (0.68)** |
| | AUPRC ↑ | 57.59 (2.10) | 75.06 (1.81) | 74.73 (1.52) | 71.62 (0.12) | 74.52 (0.83) | 74.36 (1.01) | 75.00 (1.05) | 75.17 (0.94) | **77.62 (1.43)** |

Table 6: Linear probe classification results using cross validation splits. Each split uses a different random seed for model initialization, and we report the standard deviation in parentheses. PULSE achieves the best performance across all datasets, and the results closely match those obtained with a single split.

PULSE is robust to within-dataset variability and performs consistently across datasets with diverse signal characteristics.

The semi-supervised learning results show a similar trend. In Table E, PULSE consistently outperforms all baseline methods, with the single exception of PPG at 1% labels, where it slightly underperforms TimeMAE. However, this difference is not significant since the accuracies are well within one standard deviation of each other. Importantly, these cross-validated results closely match the single split results in Table 3. This consistency across evaluation settings indicates that PULSE's representations are stable with respect to how the data is partitioned, which further strengthens our conclusion that PULSE can learn label-efficient representations that generalize across physiological datasets with very different signal and dataset characteristics.

| Dataset | | Supervised | SimCLR | TS2Vec | REBAR | PatchTST | TimeMAE | LFADS | DSVAE | SimMTM | PULSE |
|---|---|---|---|---|---|---|---|---|---|---|---|
| 1 % | HAR | 81.37 (2.88) | 72.65 (1.70) | 77.91 (1.84) | 78.75 (1.74) | 33.98 (1.96) | 81.02 (1.78) | 78.54 (1.89) | 78.28 (1.62) | 79.92 (1.62) | **84.16 (2.14)** |
| | ECG | 69.20 (2.57) | 70.03 (4.50) | 74.83 (3.97) | 71.95 (4.52) | 58.31 (5.95) | 57.94 (4.79) | 66.30 (4.50) | 71.54 (6.20) | 82.02 (2.01) | **85.74 (1.30)** |
| | PPG | 43.55 (5.75) | 37.68 (4.24) | 42.83 (5.76) | 43.33 (6.05) | 42.92 (4.48) | **44.40 (4.83)** | 39.19 (3.99) | 43.24 (5.42) | 39.51 (4.51) | 43.51 (5.81) |
| | EEG | 72.32 (1.84) | 60.27 (2.04) | 75.34 (0.74) | 74.24 (1.59) | 68.32 (1.57) | 69.10 (1.43) | 71.80 (1.81) | 73.98 (1.67) | 67.41 (1.69) | **77.95 (1.26)** |
| 5 % | HAR | 91.65 (1.35) | 82.99 (1.24) | 88.80 (1.34) | 88.64 (1.50) | 54.60 (1.89) | 91.10 (1.42) | 89.45 (1.32) | 87.38 (1.48) | 89.95 (1.33) | **92.85 (1.30)** |
| | ECG | 80.71 (3.58) | 69.64 (3.31) | 74.94 (3.46) | 71.36 (2.73) | 62.11 (6.48) | 62.37 (5.92) | 64.79 (2.57) | 73.36 (3.70) | 80.07 (2.20) | **84.95 (1.43)** |
| | PPG | 54.02 (2.24) | 42.25 (1.92) | 50.79 (2.58) | 52.74 (3.20) | 53.30 (2.43) | 45.66 (3.95) | 42.61 (2.84) | 53.75 (2.65) | 46.16 (2.86) | **54.38 (3.61)** |
| | EEG | 73.86 (2.16) | 63.38 (1.83) | 74.42 (0.50) | 73.13 (1.42) | 74.00 (1.44) | 74.76 (0.81) | 72.21 (1.75) | 72.56 (1.45) | 69.42 (1.43) | **77.87 (1.14)** |

Table 7: Semi-supervised classification accuracy for 1% and 5% of labels. Results are averaged over 5-fold cross-validation splits each with a random model initalization. Higher score is better. PULSE achieves the best performance in all settings, except for PPG at 1%, where it is within the margin of error of the top score.

## G    VISUALIZATION OF PULSE REPRESENTATIONS

We visualize the embeddings produced by PULSE using t-SNE. As shown in Figure 5, PULSE learns a representation space that effectively separates the different label classes, indicating that it captures features corresponding to clinically relevant states. In the HAR dataset, PULSE separates the representations into three major super-clusters. One cluster corresponds to laying, another merges standing and sitting activities, reflecting their similar motion patterns, and the third encompasses all walking-related activities, including walking on a flat surface, walking upstairs, and walking downstairs. In ECG, there is a clear separation between A-fib from normal rhythms. Interestingly, while normal states form multiple distinct clusters, A-fib appears as a single dominant cluster, suggesting that there are many variations of normal activity, but only one typical patterns for A-fib. Clustering in PPG is less distinct, likely due to the dataset's challenging nature and the presence of motion artifacts. Nevertheless, PULSE is able to separate stress, amusement, and meditation states, although it struggles to distinguish the baseline state. In EEG, PULSE clearly separates wake

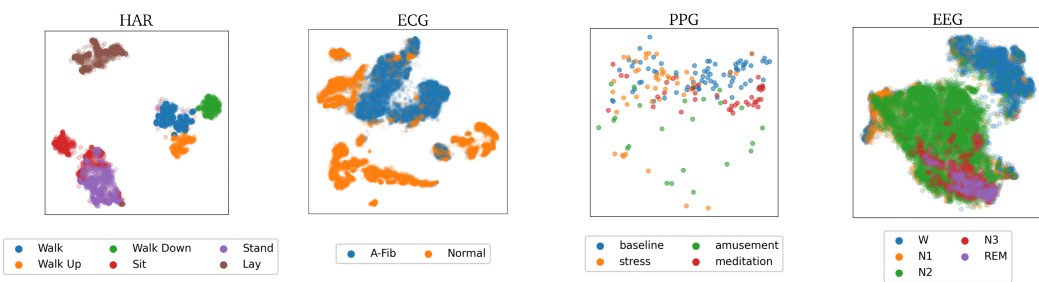

Figure 5: t-SNE visualizations of PULSE representations.

from sleep states. Furthermore, within sleep, the representation captures a continuous progression from N1 through REM states.

## H    DESCRIPTION OF MODEL TRAINING AND HYPERPARAMETERS

**Pretraining.** All models are implemented in PyTorch 2.8. We pretrain using the AdamW optimizer Loshchilov and Hutter (2017) with a One Cycle learning rate scheduler (Smith and Topin, 2019). Regularization includes gradient clipping (fixed at 5 for all models) and weight decay, which is selected through hyperparameter tuning. Models are trained for the full number of epochs, and the checkpoint with the lowest validation loss is used for evaluation.

**Linear Probe.** We evaluate the linear probe by encoding time-series samples with a frozen model and training a Logistic Regression classifier on the standardized embeddings. We use the cuML implementation (Raschka et al., 2020), a GPU-accelerated drop-in replacement for scikit-learn. Following prior work, all hyperparameters of the linear probe are fixed, including a regularization coefficient of 1, ensuring that performance differences reflect the quality of the learned representations rather than the probe itself.

**Fine-tuning.** For the fine-tuning experiments, we follow the preprocessing and dataset-splitting procedure of Zhang et al. (2022). We initialize the encoder with the best pretraining checkpoint and attach a two layer fully connected classification head with a hidden dimension of 64 and an output dimension matching the target dataset. Fine tuning is performed for 40 epochs using a batch size of 30, a learning rate of 0.0003, and a weight decay of $10^{-5}$.

**Hyperparameters.** We tune hyperparameters for each method using a random search with a budget of 30 trials over a grid of reasonable values. For trial, we uniformly sample within each of the following grids and select the best setting according to the best validation performance. Training hyperparameters are swept over epochs $[50, 100, 200]$, learning rates $[0.001, 0.0005, 0.0001]$, and weight decay $[10^{-3}, 10^{-4}, 10^{-5}]$. Model-specific hyperparameters are varied to capture differences in architecture and training objectives.

For PULSE, the model-specific hyperparameter sweep includes the initial condition encoder hyperparameters, including convolution kernel size $[3, 5, 11]$, dilation $[1, 2]$, and hidden dimensionality $[64, 128]$. For the GRU decoder, we sweep across the number of layers $[2, 3, 4]$ and hidden dimensionality $[64, 128]$. Finally, for the pseudo-pair construction, we vary the number of samples $[1, 2, 3]$. Hyperparameter grids for other baselines are included in the code repository.

## I    DISCUSSION ON PULSE'S APPROXIMATE INDEPENDENT SAMPLES.

While PULSE does not generate fully independent samples, it does retain key properties of independent samples that allow it to achieve strong performance in both synthetic and real-world evaluations. We discuss two of these properties below:

**Random Input-output pairs.** One important property of independent samples is the ability to generate random input–output pairs for the cross-reconstruction objective. These random pairings are important since they break the dependence between factors that are not shared between the elements

in the pair. PULSE retains this property by generating random input-output pairs with its pseudo-pair strategy where the input window is paired with outputs that are random crops. This allows the system representation to become invariant to initial-condition information, thereby encouraging the removal of information that is not useful for determining the underlying physiological state.

**Pseudo-pair reconstruction resembles full-sample reconstruction in the latent space.** Another important property of independent pairs is that they allow the system to train on a diverse set of latent trajectories, reducing the risk of learning system representations that overfit to noise within any single time-series window. A strong system representation should support the reconstruction of multiple time-series samples generated from that system.

During cross-reconstruction, independent samples allow us to train on a diversity of latent trajectories, helping to discourage the system representation from containing unnecessary sample-specific noise. This is different from partial reconstruction where the unmasked time points fix the latent states at intermediate positions of a trajectory. Because these unmasked latent states are fixed, they constrain the estimated latent states for the masked time points, which must maintain temporal continuity with the fixed positions. This dependence on fixed latent states limits the diversity of latent trajectories encountered during training with partial reconstruction and increases the risk of overfitting to specific samples. In contrast, independent samples allow for full-sample reconstruction where none of the latent states are fixed since reconstruction begins from newly estimate initial conditions.

PULSE is designed to emulate the full-sample reconstruction process by estimating a new initial condition for each pseudo-pair, rather than restricting it to the unmasked latent states of the original sample. Even when measurement values are identical, the resulting latent trajectories can differ depending on the initial condition estimated from different starting points within the window. This approach allows PULSE to generate more diverse latent trajectories than partial reconstruction, avoiding the constraints imposed by fixed intermediate latent states.

## J METHOD LIMITATIONS

An important limitation of our work is the gap between the practical PULSE algorithm and the theoretically ideal cross-reconstruction setting. While Theorem 1 suggests using fully independent samples to learn system representations invariant to sample-specific factors, PULSE relies on pseudo-pairs from the same sample. This may limit the model's ability to capture system dynamics, particularly when sample-specific variability is large. Future work could improve the identification of independent samples to better align practice with theory. Another limitation is the structure of our assumed generative model. Although it captures key system-level and sample-specific components, it may not fully reflect the complexity of real physiological signals. Future work in this direction could explore more flexible or data-driven generative models to improve the fidelity and expressiveness of learned system representations.

