# OpenReview forum: "Self-Supervised Dynamical System Representations for Physiological Time-Series"
_ICLR.cc/2026/Conference — Submitted to ICLR 2026_

### Official Review · Reviewer_oW81 · 2025-10-26

**Soundness:** 3
**Presentation:** 3
**Contribution:** 3
**Rating:** 6
**Confidence:** 4

**Summary:**

PULSE introduces a self-supervised pretraining method grounded in a dynamical-systems generative model.

**Strengths:**

1. Strong conceptual bridge between SSL and dynamical-systems theory.
2. Cross-reconstruction objective is original and well-motivated.
3, Empirical results show consistent gains across modalities.
4. Theoretical analysis clarifies what information is retained.

**Weaknesses:**

1. Mathematical proofs are brief, formal guarantees could be strengthened.
2. Requires grouping of “similar” time series, definition and implementation vague.
3. Limited comparison with modern contrastive MAE hybrids.
4. Does not explore scalability to high-dimensional multichannel data.

**Questions:**

1. Please provide a more detailed explanation of the method used to compute similarity between time-series samples during pretraining. Specifically, how is the similarity measure defined (e.g., Euclidean distance, cosine similarity, dynamic time warping)?
2. Compare with recent foundation-time-series models (e.g., Chronos, SleepTransformer). These models have demonstrated strong performance in various time-series tasks, and a comparison could help clarify the unique contributions and advantages of your model. Please highlight key differences in architecture, training strategies, and the specific domain(s) your model excels in.
3. To enhance the transparency of the model’s functionality, an ablation study comparing reconstruction and cross-reconstruction would be beneficial.
4. It would strengthen the paper to include a discussion on how your model performs across a variety of datasets with different characteristics.
5. Please provide further details on the training and optimization process, including hyperparameter tuning, loss functions, and any regularization techniques employed.

---

> ### Author Response · Authors · 2025-11-22
>
> ## Weakness 1
>
> Thank you for the comment. While we’re not sure if the reviewer saw this, we note here that Appendix A presents a complete proof of Theorem 1. It includes the formal definition of the reduced two-sample generative model, definitions for masked and unmasked sets, a description of Algorithm 1 from Kong and Zhang (2023), and a detailed discussion of how this algorithm can be used to prove the “if” and “only if” directions of the theorem while covering all edge cases of masking. If there are more specific critiques about which parts of this proof could be made more thorough, we would appreciate this expert insight from the reviewer.
>
> ## Weakness 2
>
> Thank you for pointing out this confusion that we did not clarify sufficiently in our original manuscript. We use the notion of “similarity” as a motivating concept to implicitly guide the extraction of system information from the data. Specifically, “similar time series” is defined as independently generated trajectories that share the same underlying system. Two time series are independent if their initial conditions, process noise, and measurement noise are independently sampled.
>
> Importantly, **our framework does not require computing an explicit similarity score.** Instead, our cross-reconstruction pre-training objective is formulated as an L2 (Euclidean) reconstruction loss in the data space. Our theory guarantees that minimizing this L2 loss implicitly recovers a representation that captures the underlying system information between similar time-series.
>
> We thank the reviewer for pointing out that this was not clear in the text. In response, **we have updated section 3.1 to include a definition of similar time-series.**
>
> ## Weakness 3
>
> We thank the reviewer for suggesting this baseline and have **added linear probe and semi-supervised learning results for SimMTM, a recent contrastive MAE hybrid method, in Appendix F** as part of the cross-validation experiments.
>
> The table below shows the linear probe results. PULSE consistently outperforms SimMTM across all datasets. Notably, PULSE achieves the largest gains over SimMTM on PPG and EEG, two noisier and more complex datasets, highlighting its ability to learn effective representations from signals with challenging properties. This demonstrates that extracting system information is highly effective across a wide range of physiological time series domains.
>
> | Dataset | Metric    | SimMTM  | PULSE   |
> |---------|----------|---------|--------|
> |         | Acc ↑    | 95.19   | **95.30** |
> | HAR     | AUROC ↑  | **99.73**  | **99.73** |
> |         | AUPRC ↑  | 98.24   | **98.99** |
> |---------|----------|---------|--------|
> |         | Acc ↑    | 53.93   | **63.87** |
> | PPG     | AUROC ↑  | 75.74   | **79.77** |
> |         | AUPRC ↑  | 51.79   | **58.43** |
> |---------|----------|---------|--------|
> |         | Acc ↑    | 82.99   | **88.01** |
> | ECG     | AUROC ↑  | 94.35   | **96.95** |
> |         | AUPRC ↑  | 90.49   | **95.32** |
> |---------|----------|---------|--------|
> |         | Acc ↑    | 80.53   | **84.25** |
> | EEG     | AUROC ↑  | 94.65   | **96.33** |
> |         | AUPRC ↑  | 75.17   | **77.62** |
> |---------|----------|---------|--------|
>
>
> ## Weakness 4
>
> Thank you for the suggestion! **We ran additional experiments on the putEMG [1] dataset, which includes high-dimensional 24-channel sEMG signals** used for a hand gesture recognition task. The dataset contains recordings from 44 subjects. We processed the data into 1-second segments and constructed a 7-class classification problem, with classes corresponding to: 1) Fist, 2) Flexion, 3) Extension, 4) Pinch-Index, 5) Pinch-Middle, 6) Pinch-Ring, and 7) Pinch-Small. We split the dataset trial-wise into 70/15/15 train/validation/test splits and averaged results over 5 random model initialization seeds.
>
> The table below shows the linear probe results for this experiment. We see that PULSE outperforms all baselines, indicating that our approach for extracting system information can also be effective in high-dimensional, multichannel data.
>
> | Method  | Accuracy ↑      | AUROC ↑        | AUPRC ↑        |
> |---------|------------------|----------------|----------------|
> | SimCLR  | 31.13 (2.65)     | 72.89 (1.11)   | 32.65 (1.44)   |
> | TS2Vec  | 61.80 (2.12)     | 89.98 (0.52)   | 68.48 (1.55)   |
> | REBAR   | 60.87 (2.30)     | 89.41 (0.23)   | 66.17 (0.25)   |
> | TimeMAE | 41.93 (1.32)     | 79.64 (0.83)   | 45.17 (1.20)   |
> | LFADS   | 55.80 (3.02)     | 85.91 (1.25)   | 59.74 (2.21)   |
> | DSVAE   | 61.33 (1.89)     | 88.93 (1.24)   | 65.80 (2.10)   |
> | SimMTM  | 60.33 (1.49)     | 89.08 (0.97)   | 65.79 (2.31)   |
> | **PULSE** | **65.27 (2.81)** | **91.34 (1.18)** | **71.43 (3.53)** |
>
> [1] putEMG—A Surface Electromyography Hand Gesture Recognition Dataset. Kaczmarek et al. 2019.
>
>
> ## Question 1
>
> See weakness 2.

---

> ### Author Response · Authors · 2025-11-22
>
> ## Question 2
>
> Great suggestion! We evaluate against the MOMENT [5] family of time-series foundation models (FMs). MOMENT models are trained across multiple datasets and domains and have been explicitly evaluated for classification tasks. The table below shows linear probe results for the small, base, and large MOMENT models which have 40M, 125M, and 385M parameters respectively.
>
>
> | Dataset | Metric     | MOMENT Small | MOMENT Base | MOMENT Large | PULSE        |
> |---------|------------|--------------|-------------|--------------|--------------|
> |         | Accuracy ↑ | 69.01        | 69.01       | 67.61        | **93.27**    |
> | **HAR** | AUROC ↑    | 92.15        | 92.48       | 91.60        | **99.42**    |
> |         | AUPRC ↑    | 78.10        | 79.62       | 77.63        | **98.10**    |
> |---------|------------|--------------|-------------|--------------|--------------|
> |         | Accuracy ↑ | **88.54**    | 88.10       | 80.27        | 87.41        |
> | **ECG** | AUROC ↑    | 96.29        | **98.32**   | 94.42        | 94.93        |
> |         | AUPRC ↑    | 96.20        | **98.15**   | 94.20        | 94.75        |
> |---------|------------|--------------|-------------|--------------|--------------|
> |         | Accuracy ↑ | 52.81        | 59.55       | 58.43        | **64.27**    |
> | **PPG** | AUROC ↑    | 77.22        | 78.29       | 73.80        | **80.29**    |
> |         | AUPRC ↑    | 54.75        | 56.59       | 51.61        | **59.89**    |
> |---------|------------|--------------|-------------|--------------|--------------|
> |         | Accuracy ↑ | 79.57        | 79.09       | 77.78        | **85.56**    |
> | **EEG** | AUROC ↑    | 94.10        | 93.59       | 93.79        | **96.17**    |
> |         | AUPRC ↑    | 67.21        | 65.87       | 64.72        | **73.82**    |
>
> **PULSE outperforms MOMENT FMs in HAR, PPG, and EEG by a significant margin.** Even though MOMENT includes HAR and EEG data in its pretraining corpus, it is not able to leverage their multi-dataset pretraining to achieve performance that is competitive with PULSE, even for in-domain generalization. For ECG, while MOMENT-Small and MOMENT-Base achieves a small improvement over PULSE, MOMENT-Large performs substantially worse. This suggests that current time-series FMs have not yet benefited from scaling laws for physiological classification tasks. The trend is consistent across HAR, ECG, and EEG, where increasing model size leads to lower performance.
>
> Although MOMENT and PULSE can both be viewed as variants of masked autoencoding, the two approaches differ in how they extract temporal and dynamical information. MOMENT models are trained with a standard MAE objective that does not explicitly target dynamical system structure and does not impose architectural constraints that encourage the learning of temporal relationships. As a result, models trained with this objective may treat patches as independent units and fail to capture important temporal dependencies.
>
> In contrast, PULSE formulates MAE training as a cross reconstruction task that directly extracts system level information and introduces architectural constraints, such as the GRU decoder, that explicitly prioritize temporal relationships. This difference in training strategy and architecture makes PULSE significantly more parameter efficient. **PULSE achieves large gains compared to MOMENT** in linear probe performance for HAR, PPG, and EEG and competitive performance in ECG **with only ~650k parameters, which is ~1% of the parameters in MOMENT-SMALL.**
>
> We did not compare against Chronos since it does not directly support classification of physiological time series out of the box. Similar to the majority of time-series FMs (e.g., TimeGPT [1], TimeLLM [2], Moirai [3], Lag-LLama[4]), Chronos focus exclusively on forecasting and is trained to directly minimize forecasting error. Adapting the features from these FMs for classification of general multivariate time series requires nontrivial method development and is therefore out of scope for this comparison.
>
> We also did not compare against SleepTransformer as an FM baseline because its pretrained weights are dataset-specific. It is trained only on the SHHS dataset, which contains only sleep data. Consequently, its pretrained features are unlikely to transfer well across the different physiological domains considered in our evaluation.
>
> [1] TimeGPT-1. Garza et al. 2024
>
> [2] Time-LLM: Time Series Forecasting by Reprogramming Large Language Models. Jin et al. 2024
>
> [3] Unified Training of Universal Time Series Forecasting Transformers. Woo et al. 2024
>
> [4] Lag-Llama: Towards Foundation Models for Probabilistic Time Series Forecasting. Rasul et al. 2024.
>
> [5] MOMENT: A Family of Open Time-series Foundation Models. Goswami et al. 2024

---

> ### Author Response · Authors · 2025-11-22
>
> ## Question 3
>
> We agree! **In fact, this baseline is already included in our original submission in Table 5 under the condition “w/o Sample $t_0$”.** When the initial condition is not sampled and  is always taken from the first time-step, our pseudo-pair strategy collapses to using the same time-series as bot the input and the target i.e. $(Y_i,Y_i)$, which corresponds exactly to direct reconstruction. We thank the reviewer for noting this is unclear and have updated the ablation conditions in Table 5 and added clarifications in Section 5.4 to make explicit that this setting represents direct reconstruction.
>
>
> ## Question 4
>
> Great point! We agree that it's important to highlight that our model is effective across a broad range of datasets with very different characteristics. This is precisely why we evaluate four physiological datasets spanning diverse dataset characteristics and physiological semantics. These datasets include different sensor types to record distinct physiological processes from various locations on the body. Each of these datasets are described in detail in Appendix D.
>
> Overall, we find that **PULSE can effectively represent a broad range of physiological semantics.** Our evaluations cover human movement (HAR), the cardiovascular system (PPG, ECG), and brain activity (EEG). Each signal has very distinct properties. For instance, ECG exhibits sharp quasi-periodic peaks, whereas HAR does not. EEG signals are frequency focused, and PPG signals include many recording artifacts. Since PULSE consistently outperforms baselines methods, this demonstrates that our dynamical systems formulation is capable of learning meaningful representations from signals originating from very different physiological processes.
>
> These different physiological semantics lead to different dataset properties. For instance, HAR has a short context length (2.56s) since activity can change on a moment by moment basis while EEG has a longer context length (30s) which is defined to be a unit of sleep by the American Academy of Sleep Medicine, which considers EEG activity to be similar during one period. Moreover, HAR requires a multi-channel setup to record different axes of motion while PPG only requires a single channel to optically record blood activity. These differences in sensor configurations further highlight PULSE’s ability to learn meaningful representations across different dataset properties, handling both short and long contexts as well as single- and multi-channel recordings.
>
> Thank you for this suggestion! **We added a paragraph in Section 5.2 that incorporates this discussion.** In particular, we emphasize that PULSE can achieve strong performance across diverse physiological processes with very different signal properties and dataset characteristics.
>
> ## Question 5
>
> We agree that providing additional training and optimization details would improve the clarity of our work. In response, **we have added Appendix H, which describes the full training setup for pretraining, linear probing, and fine-tuning.** This includes our hyperparameter tuning procedure as well as the regularization techniques used during pretraining (weight decay, gradient clipping, and learning-rate scheduling). We note that the PULSE pretraining loss function is fully defined in Equation 3 (Section 3.2), where we also explain the motivation behind this formulation.

---

> > ### Comment · Reviewer_oW81 · 2025-11-23
> >
> > Thank you for the response. The underlying concept is nice, although neural spiking is, to me, inherently highly variable. Without labels, the model cannot determine whether two samples truly originate from the same system, and the reliability of the pseudo-pairs entirely on the strong assumption of intra-sample state stability. Since my score is not negative, I will keep it.

---

> > > ### Author Response · Authors · 2025-11-24
> > >
> > > We apologize for any misunderstanding about our datasets and the goals of our work. **There is no neural spiking data in our analysis.** Our two neuro-related datasets include EEG and the newly added surface-EMG (sEMG), which both reflect aggregated electrical activity from millions of neurons of the brain and muscle fibers respectively. Neither includes signals from individual neuron spikes. Moreover, **our goal is NOT to explicitly verify if “samples truly originate from the same system”. Instead, our goal is to learn transferable representations for physiological signals** in datasets without labels, which follows the existing time-series SSL literature. While we use dynamical systems to inspire our pretraining method, we do not assess the recovery of the true systems. This recovery is not aligned with the goals of SSL, which focuses on learning representations that transfer effectively, since perfectly recovering the underlying system does not guarantee representation transferability.
> > >
> > > To this end, we validate the “reliability of the pseudo-pairs” by extensively evaluating 4 real datasets with distinct properties and noise characteristics. This helps **showcase that our pseudo-pair strategy is broadly reliable, even when perfect intra-sample state variability may not hold.** Based on your excellent feedback, we have further strengthened this claim by adding **a new sEMG dataset, two new baselines** (i.e. SimMTM, MOMENT), **additional baselines in the ablation study,** and **new cross-validation experiments** where we demonstrate that PULSE consistently achieves the best performance. Moreover, we address earlier concerns about the text’s clarity by adding a new discussion of performance across different dataset characteristics, providing an explicit definition of similar samples, and including a new section detailing our training setup.
> > >
> > > As such, during our rebuttal, we believe we have addressed this concern, as well as your prior concerns on missing baselines and our work’s clarity. **Could you please clarify your newly raised concerns and let us know what additional evidence or experiments you would need to raise your score?** For example, what specific aspects of data variability or pseudo-pair reliability are not addressed by our current experiments, and what is the source of confusion regarding our evaluation goals?

---

> > > > ### Comment · Reviewer_oW81 · 2025-11-27
> > > >
> > > > I appreciate the additional datasets and baselines included during the rebuttal. My point is that, even for physiological signals such as EEG or sEMG, the intra-sample dynamics can still vary substantially due to nonstationarity, noise, and context changes. In such settings, the pseudo-pair assumption may not reliably reflect a stable state or shared underlying dynamics. While your rebuttal adds more datasets and baselines, it does not fully address the core conceptual issue: whether the pseudo-pairing mechanism remains valid when the assumptions of local state stability or temporal coherence are weakened.

---

> > > > > ### Author Response · Authors · 2025-11-27
> > > > >
> > > > > Thank you for elaborating on your concern. Our rebuttal already includes experiments on real-world data where the assumption of intra-sample temporal coherence may not necessarily hold perfectly, yet PULSE continues to outperform all baselines. For example, EEG is highly non-stationary due to rapid transitions between metastable neural states [1], and even 0.25-second windows are treated as only quasi-stationary [2]. Despite this, we show that PULSE’s time-varying system variables allow it to successfully recover the underlying physiological state even in this nonstationary setting, from 30-second EEG windows.
> > > > >
> > > > > Could you clarify what additional experiments you would consider as evidence that the “pseudo-pairing mechanism remains valid when assumptions of local state stability or temporal coherence are further weakened”?
> > > > >
> > > > > [1] Everything you wanted to ask about EEG but were afraid to get the right answer. Klonowski  2009.
> > > > >
> > > > > [2] Neuronal coordination in the brain: A signal processing perspective. Kaplan 2005.

---

### Official Review · Reviewer_5J2C · 2025-10-26

**Soundness:** 2
**Presentation:** 3
**Contribution:** 3
**Rating:** 4
**Confidence:** 3

**Summary:**

The paper addresses SSL for physiological time series, claiming that the standard methods: CL, MAE, and standard SVAEs, fail to isolate shared "system" dynamics from sample specific noise. It proposes PULSE, a framework based on a hierarchical dynamical systems view. The core idea is a pretraining objective that reconstructs a window from a randomly sampled start time within the same window, aiming to learn representations invariant to initial conditions. Experiments across synthetic and four real datasets (HAR, ECG, PPG, EEG) demonstrate PULSE often outperforms standard baselines.

**Strengths:**

1. The motivation is sound. Generic SSL heuristics are generally not suited for physiological signals where temporal structure is key. Framing this through dynamical systems is appropriate.
2. The paper provides a thorough comparison to other baselines across different algorithmic families.
3. The paper shows consistent performance improvements across diverse tasks especially on complex signals like EEG/ECG.

**Weaknesses:**

1. There appears to be a disconnect between the theoretical justification and the practical implementation. Theorem 1 states that full-sample masking (cross-reconstruction between independent samples) is required to isolate system parameters. However, PULSE uses partial window reconstruction from the same sample. The proof in appendix A indicates that partial masking fails to isolate and retains sample-specific information. The paper frames this as an approximation, but it seems to violate the conditions of the theory used to justify it.
2. The paper positions PULSE as a principled alternative to heuristic methods. However, it relies on its own set of design choices, such as the specific uniform sampling of t_0. It is unclear how this is fundamentally different from applying a random temporal cropping augmentation to a standard SVAE, a type of heuristic the paper initially critiques in the introduction.
3. The framework aims to separate "system" information from "initial conditions." However, since both encoders observe the same sample Y_i, and the system representation includes time-varying components $\tilde{\theta}_{i,t_k}$ derived from that same input, there is a risk of information leakage. The encoder could potentially copy local signal values directly into these time-varying parameters, allowing the decoder to reconstruct the signal without learning true underlying dynamics.
4. The hierarchical model assumes a clean separation between shared "system parameters" and unique "initial conditions," which relies on the signal being stationary within the window W. In many physiological signals, this assumption may not hold. If W is small, there is a risk the "system" encoder might overfit to local, non-stationary patterns rather than true underlying dynamics. So selecting the right W is also somewhat of a heuristic like point 3 above.

**Questions:**

Please address the weaknesses section above.

---

> ### Author Response · Authors · 2025-11-22
>
> ## Weakness 1
>
> Thank you for the comment. We are grateful for the chance to clarify here. While PULSE does not generate fully independent samples, it does retain key properties of independent samples that allow it to achieve strong performance in both synthetic and real-world evaluations. We discuss two of these properties below:
>
>
> * **Random Input-output pairs.**
>
>     * **What property of independence is retained?** One important property of independent samples is the ability to generate random input–output pairs for the cross-reconstruction objective. These random pairings are important since they break the dependence between factors that are not shared between the elements in the pair.
>
>      * **How does PULSE retain this property?** PULSE retains this property by generating random input-output pairs with its pseudo-pair strategy where the input window is paired with outputs that are random crops. This allows the system representation to become invariant to initial-condition information, thereby encouraging the removal of information that is not useful for determining the underlying physiological state.
>
> * **Pseudo-Pair’s produce more diverse latent trajectories than direct reconstruction.**
>     * **What property of independence is retained?** Another important property of independent pairs is that they allow the system to train on a diverse set of latent trajectories, reducing the risk of learning system representations that overfit to noise within any single time-series window. Our guiding principle is that a strong system representation should support the reconstruction of multiple time-series samples generated from that system.
>
>         \
>         During cross-reconstruction, independent samples allow us to train on a diversity of latent trajectories, helping to discourage the system representation from containing unnecessary sample-specific information.
>
>         \
>         This is different from partial reconstruction where the unmasked time points fix the latent states at intermediate positions of a trajectory. Because these unmasked latent states are fixed, they constrain the estimated latent states for the masked time points, which must maintain temporal continuity with the fixed positions. This dependence on fixed latent states limits the diversity of latent trajectories encountered during training with partial reconstruction and increases the risk of overfitting to specific samples.
>
>         \
>         In contrast, **independent samples allow for full-sample reconstruction** where none of the latent states are fixed since reconstruction begins from newly estimated initial conditions.
>
>     * **How does PULSE retain this property?** PULSE is designed to emulate the full-sample reconstruction process by estimating a new initial condition for each pseudo-pair, rather than restricting it to the unmasked latent states of the original sample. Even when measurement values are identical, the resulting latent trajectories can differ depending on the initial condition estimated from different starting points within the window. This approach allows PULSE to generate more diverse latent trajectories than strict partial reconstruction, avoiding the constraints imposed by fixed intermediate latent states.
>
> In response to your comment, we agree that the gap between the practical PULSE algorithm and the theoretically ideal cross-reconstruction setting is an important limitation. As a result, **we have added Appendix J to discuss the limitations of our work** and acknowledge this gap between theory and implementation. Moreover, **we add Appendix I to discuss how PULSE approximates the properties of independent samples.**

---

> ### Author Response · Authors · 2025-11-22
>
> ## Weakness 2
>
> Thank you for the comment, which highlights a discussion that we did not make clear enough in the previous draft. We will clarify how PULSE differs from existing literature in a few ways below and comment on how our insufficient descriptions (which we will update) led to this confusion:
>
> 1. **Distinguishing PULSE from Heuristic Methods.** We thank the reviewer for pointing out that our description of heuristic methods in the main text may be interpreted ambiguously. In our work, we distinguish between *information-driven* design choices, which are intended to shape the type of information extracted by the model, and *implementation-driven* design choices, which do not aim to influence the extracted information but are necessary to implement a specific method.
>
>     \
>     We define heuristics as information-driven model design choices that are empirically derived without providing theoretical justification. Many CL and MAE strategies are considered heuristic because the information they capture depends on design choices made to empirically improve downstream performance. Often, these choices do not explicitly state the assumptions about the data, specify which sources of information are captured, or explain how the method targets those sources.
>
>     \
>     PULSE differs from heuristic SSL because its information-driven design choices are based on a data-generating model that makes the assumed data structure explicit, and specifies which sources of information should be recovered and which should be ignored. This explicit framework is beneficial since it allows us to formally analyze our method and provide theoretical justification for how PULSE can target the desired sources of information.
>
>     \
>     Moreover, the choice to sample t_0 uniformly is implementation-driven and therefore is not considered a type of heuristic that we criticize in our paper since it is not intended to influence the type of information that is targeted or extracted during pretraining. Accordingly, **we disagree that PULSE’s implementation-driven design choices make it a heuristic method**. However, we agree that this discussion can be made more clear in the text, and **we update the introduction to improve its clarity that “heuristics” refers to information-driven model design choices.**
>
>      \
>   2. **Distinguishing PULSE from SVAEs with random temporal crop.** Our main critique of SVAEs is not that they are heuristic, but that **SVAEs use a direct reconstruction objective**. In contrast, **PULSE uses a cross-reconstruction objective**.  This distinction is important because direct reconstruction jointly observes signal and noise and uses fixed input-output pairs. This makes it difficult to separate the signal from noise and limits the diversity of latent trajectories a system representation can reconstruct. Cross-reconstruction breaks this dependency by forming random input-output pairs, which increases the diversity of latent trajectories that a system representation must be robust to. This allows PULSE to learn a latent space that captures information shared between pairs while discarding information that is not shared.
>
> ## Weakness 3
>
> We agree that there is a potential risk of information leakage from the initial condition into the system representation. In our work, we employ the following two strategies to mitigate this risk, both based on limiting the expressivity of the time-varying components.
>
> 1. First, we constrain the dimensionality of the time-varying parameters to a single dimension. This ensures that the time-varying component alone does not have sufficient capacity to represent the full diversity of initial conditions in the data, particularly for multi-channel observations.
>
> 2. Second, we limit the temporal variability of $\theta_{i,k}$​ by reducing its ability to change between timesteps. Specifically, we apply adaptive max pooling across the time dimension for the time-varying components and assign the pooled values to the corresponding variables, effectively sharing the time-varying components across multiple consecutive timesteps.
>
> Together, these two strategies encourage a clearer separation between system information and initial conditions. We thank the reviewer for the comment and **have updated Section 3.2 to include this discussion.**

---

> ### Author Response · Authors · 2025-11-22
>
> ## Weakness 4
>
> The choice of window size is not a hyperparameter of the model, and is not the type of heuristic we critique in our paper. We primarily criticize heuristic choices intended to affect the type of information extracted by the model.
>
> Conversely, W determines the temporal scale at which the pretraining objective is computed, it does not dictate which sources of information the model should extract. For the benchmark datasets we consider, W’s are determined based on domain knowledge of how a single unit of a particular semantic class is defined and it is generally assumed that the properties of these semantic classes are approximately stationary within each window. For instance, in the Sleep-EDF dataset, the American Academy of Sleep Medicine defines the sleep period length as 30-second windows and considers EEG activity to be similar during one period [1]. Thus for EEG in the sleep staging task, each 30-second window is assigned a single label. While changing W may affect the scale over which information is extracted, it does not change the type of information that the model is intended to capture.
>
> [1] The AASM manual for the scoring of sleep and associated events: rules, terminology, and technical specification. Iber 2007.

---

> > ### Comment · Reviewer_5J2C · 2025-11-24
> >
> > I thank the authors for their detailed response.
> >
> > However, my primary concerns regarding the paper's theoretical positioning remain unresolved. While I appreciate the transparency in adding Appendix J to acknowledge the gap between the ideal cross-reconstruction required by Theorem 1 and the practical pseudo-pair implementation, simply acknowledging this discrepancy does not resolve the scientific inconsistency: the proposed algorithm does not satisfy the theoretical conditions used to justify its "principled" nature.
> >
> > Furthermore, I find the distinction between information-driven and implementation-driven heuristics unconvincing. The random sampling of $t_0$ is functionally identical to the random cropping augmentations used in the heuristic methods criticized in the introduction. As the method effectively amounts to an SVAE with data augmentation, the theoretical framing remains unsupported by the implementation.
> >
> > For these reasons, I'd like to maintain my original score.

---

> > > ### Author Response · Authors · 2025-11-26
> > >
> > > Thank you for elaborating your concerns, and we aim to address each point below. Please let us know what additional questions you have to help alleviate any concerns.
> > >
> > > # 1. Theoretical Positioning
> > > We agree that our positioning can be improved and have softened our language on our theory contribution in the introduction (line 105-106). **To further discuss this idea though, gaps between idealized theoretical settings and practical implementations are common in the representation learning literature, often necessary for tractable learning, and are generally not viewed as issues of “scientific consistency”.** For example, the Kalman VAE [1] assumes that the latent structure follows Markovian, locally linear dynamics in order to have a tractable approximate posterior, but this data assumption may not hold true in many time-series because the true data-generating processes can exhibit strongly nonlinear behavior or long-term dependencies. It is common for VAE methods [2,3,4] to ignore such discrepancies in order to allow for practical implementations.
> > >
> > > **These theoretical-to-practical approximations are widely used and effective**. Similarly, based on your earlier feedback, we outline in the **new Appendix I** that PULSE’s pseudo-pairs retain key properties of independent samples, and our theory guarantees that greater independence leads to more accurate recovery of system information. While PULSE does not satisfy every theoretical condition of the data exactly, this does not undermine its effectiveness for learning transferable SSL representations, which is the primary goal of our work. **Our theoretical framing provides an explanation for its effectiveness while suggesting directions to improve future SSL methods.**
> > >
> > > # 2. Comparison to SVAE
> > > Moreover, **PULSE does not “effectively amount to an SVAE with data augmentation”.** PULSE learns a vector embedding useful for SSL transferability instead of the latent distribution expected from VAE methods. Additionally, PULSE uses a cross-reconstruction learning objective to learn shared information from the original signal and a randomly cropped version, thereby discarding noise from the initial condition. This differs from SVAEs that only learn to reconstruct a cropped signal from itself and may discard unknown components of the signal. Our empirical results demonstrate that this distinction is important with PULSE achieving significantly stronger performance.
> > >
> > > # 3. Random Crop being not heuristic
> > > The random crop we argue is not heuristic because it’s motivated by our data-generating framework which identifies that initial condition information is sample-specific and should be removed.
> > >
> > > [1] A Disentangled Recognition and Nonlinear Dynamics Model for Unsupervised Learning. Fraccaro et al. 2017
> > >
> > > [2] Generative Modeling of Regular and Irregular Time Series Data via Koopman VAEs. Naiman et al. 2024
> > >
> > > [3] Probabilistic forecasting with VAR-VAE: Advancing time series forecasting under uncertainty. Leushuis 2025.
> > >
> > > [4] Markovian Gaussian Process Variational Autoencoders. Zhu et al. 2023

---

### Official Review · Reviewer_dmUK · 2025-10-30

**Soundness:** 2
**Presentation:** 2
**Contribution:** 2
**Rating:** 2
**Confidence:** 4

**Summary:**

This paper introduces a self-supervised framework for learning embeddings of physiological time series based on dynamical system modeling. By considering dynamical system modeling, the authors propose a new pre-training method to filter out irrelevant noise while creating embeddings that account for the physiological state. Namely, the embedding intends to represent the underlying dynamical systems  by cross-estimation rather than the observed trajectories. This pre-training framework intends to overcome limitations of current methods based on contrastive learning or variational encoding. Incrementally to previous work, the authors established a theorem ensuring the recovery of dynamical systems. In the experimental section, the authors explore the validity of the established theorem in degraded situations and evaluate their representations' performance in various ML tasks, including classification and transfer learning.

**Strengths:**

- The paper is incrementally written, making it easy to follow from start to end. The cross-reconstruction and its relaxation (PULSE) are well motivated and described. Theoretical estimation guarantees further back it.
- The paper benefits from an extensive experimental study, including analyzing the theorem validity on synthetic data, performance evaluation on real-world data for various ML tasks, and an ablation study.

**Weaknesses:**

- The estimation of dynamical systems from data is also known as dynamical mode decomposition; the authors should also position their work with regard to this literature.
- The “Time-series Dataset Generative Model” (section 3.1) is not easy to follow while central to the paper. I invite the authors to revise this paragraph; they can, for instance, make better use of Figure 2.
- The experiments seem to be run on a single train-validation-test split and a single network initialization. To ensure the robustness of the experimental results, I invite the authors to address this issue.
- As the classification experiment ends with a probe linear estimator, a 2D-visual representation of the embedding spaces would be interesting to add in the appendix to see if clusters of dynamical systems can represent the different classes.
- Format issue: In assumption 1, line 286, the abbreviation DAG is not defined.

**Questions:**

- Regarding practical settings, a point that I found unclear is the passage from latent space to observable space ($g_y$). How is it implemented in practice? Are its parameters considered in the DS embedding ($\Theta$)?
- Can the authors address my concern regarding the robustness of experimental results described in the weaknesses?
- An important baseline to compare to in the experiment would be the pretaining that considers the cross-reconstruction strategy with randomly selected pairs of subsequences. Can this baseline be added to the classification or ablation study?

---

> ### Author Response · Authors · 2025-11-22
>
> ## Weakness 1
> Thank you for raising this concern and for the opportunity to clarify our thoughts here. We respectfully disagree that our work is significantly related to dynamical mode decomposition (DMD). While both DMD and time series SSL aim to extract features from temporal data, the two literatures are largely distinct in both their goals and methodological assumptions.
> DMD refers to a specific family of algorithms for identifying low dimensional linear representations and coherent dynamical modes of a system. Rather than being a general term for estimation of dynamical systems from data, the central goal of the specific DMD literature is to recover modes that exhibit predefined temporal patterns such as damped or driven sinusoids. Consequently, DMD is a specific methodology that assumes linearity and modal decomposability, and is typically evaluated on reconstruction and forecasting accuracy in controlled or simulated dynamical systems, and on the interpretability of its eigenvalues and eigenmodes [1,2]. Crucially, **DMD outputs are not designed for transfer and it is not necessarily obvious how to use them for downstream tasks such as classification.**
>
> In contrast, the time series SSL literature focuses on learning general purpose representations that are optimized for transferability across diverse tasks and datasets, without assuming linearity or modal decomposability. SSL embeddings aim to capture high level semantic structure (e.g., distinguishing healthy vs. diseased patterns) rather than predetermined temporal modes, and they do not assume that the underlying system admits a modal decomposition. Their effectiveness is therefore evaluated through downstream performance on tasks such as linear probing, semi-supervised learning, and transfer learning [3,4], which reflect real world clinical applications.
>
> **Our work aligns much more closely in goals and methodological approaches with the time-series SSL literature** because our focus is on transferable representation learning rather than modal decomposition and interpretability. For these reasons, while both DMD and PULSE are motivated by a dynamical systems framework, we believe that DMD does not serve as a valuable alternative or competing framework for our setting, since its assumptions, objectives, and evaluation criteria differ substantially from those of the SSL literature.
>
> [1]  Variable projection methods for an optimized dynamic mode decomposition. Askham et al. 2017
>
> [2] On Dynamic Mode Decomposition: Theory and Applications. Tu et al. 2013
>
> [3] Self-Supervised Contrastive Pre-Training For Time Series via Time-Frequency Consistency. Zhang et al. 2022
>
> [4] Self-Supervised Learning for Time Series Analysis: Taxonomy, Progress, and Prospects. Zhang et al. 2022
>
>
> ## Weakness 2
> Thank you for pointing this out. We have revised Section 3.1 in response to your comment and strengthened the connection between the text and its corresponding figure. In the updated version, we explicitly state that the joint distribution in Equation 1 is illustrated by the graphical model in Figure 2. We also clarify the notation for the index sets and provide precise definitions for each probability density. Finally, we include a definition of similar time series based on the graphical model to make our notion of similarity explicit.

---

> ### Author Response · Authors · 2025-11-22
>
> ## Weakness 3
>
> **We apologize for the misunderstanding, but this is an incorrect interpretation of what we are reporting. In our original submission, we do not report results for a single network initialization from a single train-val-test split.** Instead, we reported results for a single split averaged over multiple seeds for all experiments. For linear probe, we average over 5 random network initializations. For semisupervised learning, we mention on line 471, that we average over five random label subsets for each of the five model initialization from the linear probe experiments, results in 25 random seeds. For transfer learning, we mention on line 500 that we average over five random seeds for both model initialization and fine-tuning. However, based on your comment, we realized this was not clearly stated for the linear probe experiment. In response, we have updated Section 5.1 to clarify that linear probe results are averaged over 5 random network initializations.
>
> Although this is different from the standard cross-validation (CV) setup, many prior works in time-series SSL also use fixed splits [1,2,3,4]. This is because while cross-validation (CV) is one technique to assess model generalization, it is valid to assess model generalization by demonstrating consistent performance across multiple distinct datasets. In our experiments, we take this multi-dataset approach since it allows us to directly compare our baselines to prior studies by using the same fixed splits as earlier work, while also reducing the risk of underreporting baselines due to implementation errors.
>
> We do agree that combining both CV with a multi-dataset approach can strengthen our results, and in response **we have added Appendix F, which reports a 5-fold cross validation experiment for each dataset.** We find that the results are highly consistent with the single split results, indicating that our findings are stable and not sensitive to the particular data split. Interestingly, when averaging across different dataset splits, PULSE achieves the best performance for HAR even though it was not the top performer in the single-split setting. This further supports that PULSE is robust to within-dataset variability and can consistently perform across datasets with diverse signal characteristics.
>
> [1] REBAR: Retrieval-Based Reconstruction for Time-series Contrastive Learning. Xu et al.
>
> [2] Self-Supervised Contrastive Pre-Training For Time Series via Time-Frequency Consistency. Zhang et al. 2022
>
> [3] SimMTM: A Simple Pre-Training Framework for Masked Time-Series Modeling. Dong et al. 2023
>
> [4] TimeMAE: Self-Supervised Representations of Time Series with Decoupled Masked Autoencoders. Cheng et al. 2023
>
> ## Weakness 4
>
> Thank you for this suggestion! **We have added Appendix G with t-SNE visualizations** of the embedding spaces. We see that PULSE embeddings naturally cluster according to the true label classes, indicating that system information often aligns with the underlying semantic classes for physiological time-series.
>
> ## Weakness 5
> DAG stands for **Directed Acyclic Graph**, and we have updated the text to include this definition at first use.
>
> ## Question 1
>
> The mapping $g_y$ is defined in lines 246-247 as a linear projection layer. Importantly, as shown in equation 2, $\Theta$ does not include parameters for $g_y$ and only includes parameters for $g_x$. The DS embedding $\Theta$ includes only the pooled output of the dilated convolution. This design choice is motivated by the SSM formulation, where the parameters of the observation function are not considered part of the underlying dynamics, reflecting the idea that how a process is measured is separate from the dynamics of the process itself. **We have updated the text in section 3.2 to include this clarification.**
>
> ## Question 2
>
> See response in weakness 3
>
> ## Question 3
>
> Great point! We agree that cross-reconstruction with randomly selected pairs would be an interesting baseline for the ablation study. **We have updated Section 5.4 and Table 5 to include this baseline**. As expected, this approach leads to degraded performance across all datasets, highlighting the effectiveness of PULSE’s pseudo-pair strategy.

---

> > ### Comment · Reviewer_dmUK · 2025-11-26
> >
> > I thank the authors for addressing my concerns.
> >
> > I appreciate the clarifications added to the revised manuscript, as well as the additional experiments. The authors have resolved my concerns regarding the experimental protocol by incorporating cross-validation and by adding a baseline to assess the validity of the pseudo-pair strategy. I also appreciate the detailed argument distinguishing their work from the DMD literature. I agree that DMD and SSL differ significantly in goals and evaluation criteria. I apologize for the misunderstanding. To clarify, my initial remark was not intended to position DMD as a competing method, but rather to acknowledge its relevance as a classical approach to low-rank dynamical system identification.
> >
> > Overall, the revisions address most of my concerns regarding clarity and experimental protocol. I will raise my score accordingly.

---

### Author Response · Authors · 2025-11-22
**Summary of Revisions in Response to Reviewer Feedback**

We thank the reviewers for highlighting that our work is well-motivated (dmUK, 5j2C, oW81), has thorough experimental comparisons (dmUK, oW81), and strong empirical results (5j2C, oW81). We’ve made several updates to the manuscript (denoted in orange text) that we hope address the reviewers’ critiques and uploaded a new draft with all the changes. Here, we highlight the main updates:

**1. Cross-validation experiments with additional baselines.**

Multiple reviewers requested additional experiments. In response, we added Appendix F, which includes cross-validation results for both linear probe (Table 6) and semi-supervised learning (Table 7). We also include SimMTM as a baseline, a recent competitive contrastive-MAE hybrid approach. Overall, the cross-validated results closely match the single-split results, which further supports our original observations.

Interestingly, when averaging over cross-validation folds, PULSE consistently outperforms all baselines in both linear probe and semi-supervised settings for all datasets, even though the originally reported HAR results slightly underperform REBAR for linear probe. This indicates that PULSE is robust to within-dataset variation and further supports the idea that explicitly capturing system information is a general principle for learning effective representations from a wide range of physiological signals.

**2. Improved clarity of important concepts and additional Training details.**

Multiple reviewers had critiques that asked for more discussion or represented misunderstandings that required us to revise the text to make things more clear..

In response, we revise the text to clarify our use of certain concepts that may be unclear or interpreted ambiguously. Specifically, we revised the introduction to make it explicit that “heuristics” refers to modeling choices aimed at influencing the type of information extracted by an encoder. We also update section 3.1 to make better use of Figure 2 and add a definition for our notion of “similar time-series”. Furthermore, we added Appendix I, which includes a discussion on how PULSE’s approximately independent samples satisfy key properties of fully independent samples required by the theorem.

We also expanded the text to include more details about our training and optimization procedure. In Section 3.2, we add a paragraph that describes two approaches used to regularize the time-varying variables and prevent trivial system representations. In section 5.1, we clarify that we average over 5 random seeds for model initialization for linear probe. We also add Appendix H and provide full training details for pretraining, linear probe evaluation, and fine-tuning, as well as our hyperparameter optimization procedures.


**3. Additional baselines in ablation study.**

Several reviewers offered helpful suggestions for improving the ablation study, including requests for additional experiments and clearer explanations in the text. For experiments, we include “cross-reconstruction with random pairs” as an additional ablated baseline, and demonstrate that this setting significantly underperforms the unablated PULSE, further supporting our model design decisions. We also improve the clarity of the text in section 5.4 to be more explicit that one of the ablation settings represents direct reconstruction.
___

We thank you all for your valuable feedback and the opportunity here to clarify critical comments that were based on misunderstandings fro the writing in our original draft. Our revised paper is significantly improved because of your comments.

---

### Author Response · Authors · 2025-12-03

Dear Area Chair,

We’d like to thank all of the reviewers for taking the time and effort to provide such high quality feedback. We published our responses early (11/21), which allowed us to have a highly productive discussion period. Although all of our reviewers were able to respond before the discussion freeze, we were unable to finish our discussions to reach conclusions for 2 reviewers (5J2C, oW81). **However, Reviewer dmUK did confirm that our rebuttal was able to resolve all of their concerns and had raised their score to a 6, such that we had an average 5.33 scoring (6 oW81, 4 5J2C, 6 dmUK).**

Since the quality and clarity of our work has improved significantly with the scores trending positively throughout our rebuttals, we strongly believe that Reviewers oW81 and 5J2C would have also raised their score given a full discussion period. To assist with the AC decision, we summarize the outstanding concerns and discussions with Reviewers oW81 and 5J2C below:

1. Reviewer oW81’s remaining concern is about the validity of our pseudo-pairing strategy when applied to real-world data with nonstationary dynamics, noting that “the assumption of intra sample dynamics may be invalid when local state stability is weakened” (oW81). We address this concern in theory and experiments, confirming that our pseudo-pair strategy is reliable across a diversity of nonstationary dynamics.
\
\
    **Our theory can extend to non-stationary dynamics.** Under Assumption 1, system information may vary over time as long as the underlying time-varying system parameters admit a DAG factorization. To extract system information from windows with nonstationary dynamics, PULSE uses time-varying system variables as described in section 3.2.
\
\
    Our extensive empirical results validate PULSE’s pseudo-pair strategy, demonstrating that it consistently outperforms competitive SSL baselines across diverse datasets, each exhibiting distinct forms of nonstationarity. In response to reviewer feedback, our rebuttal further assesses the validity of the pseudo-pairs by showing SOTA performance on **a new sEMG dataset, new cross-validation experiment,** and against **two new baselines** (SimMTM, MOMENT). Moreover, **we update the ablation study in section 5.4 to include new ablation baselines, explicitly demonstrating the importance of time-varying parameters and the pseudo-pair strategy**, with both ablations producing the largest drop in transfer performance. These experiments further strengthen our claim that PULSE's pseudo-pair strategy is reliable even when theoretical data assumptions may not be perfectly satisfied.
\
\
2. Reviewer 5J2C’s remaining concern is the theoretical positioning, specifically regarding the gap between the fully independent samples assumed in Theorem 1 and the approximately-independent samples used by PULSE. However, such gaps between theoretical data-assumptions and actual data-properties are common in the representation learning literature and do not undermine PULSE’s empirical effectiveness for consistently learning transferable representations in physiological time-series, which is the main goal of our work.
\
\
    We directly address this concern in our rebuttal by **softening our theoretical contribution** in the introduction and **adding two discussions that strengthen the connection between theory and PULSE**, with Appendix I explaining the independent-sample properties PULSE satisfies and Appendix J clarifying the gap between theory and practice. Our theoretical framework is a strength of our approach, as it clearly specifies the assumptions about the data-generating process for an ideal setting, highlights which assumptions may not hold fully in practice, and points to directions for future improvement by closing the gap between theory and practice.
\
\
3. Reviewer 5J2C’s secondary remaining concern is that PULSE is identical to an SVAE. We address this concern by emphasizing that PULSE’s cross-reconstruction learning objective is distinct from SVAE’s direct reconstruction learning objective. Specifically, our cross-reconstruction explicitly discards information about the initial condition, whereas direct-reconstruction may discard unknown signal components. Our ablation study shows that this distinction is important, as direct reconstruction consistently produces worse transfer performance compared to PULSE’s cross-reconstruction with pseudo-pairs. **Our approach is “original” (oW81), as no other SVAE methods currently explore cross-reconstruction in their learning objective.**

---

### Meta-Review · Area_Chair_8Lvv · 2025-12-14

**Summary:**

The submission studied self-supervised dynamic representation problem by delivering a pre-trained model across multiple time-series. Reviewer appreciated the motivation of the approach, but some concerns regarding the theoretical analysis and evaluations (comparisons with stoa methods, configurations of evaluations).

**Reviewer Concerns:**

The major concerns raised by multiple reviewers, including the theoretical analysis, and justification of splitting, are not well addressed. In terms of the theoretical analysis, which is one of the main contributions in the submission, is pointed out by reviewer to be not solid,  and also admitted by the authors. The concern of evaluation of data splitting is also not well addressed. The authors rebuttal in a way that using one data split but with multiple seeds, which to the AC, is not solid.

**Reviewer Scores:**

reviewer Dmuk provided reasonable comments, but the comments are not well addressed. thus the AC believes that the reviewer is less likely to change the score.

---

### Decision · Program_Chairs · 2026-01-26

Reject